

# Climatology of the terms and variables of transformed Eulerian-mean (TEM) equations from multiple reanalyses: MERRA-2, JRA-55, ERA-Interim, and CFSR

Masatomo Fujiwara[1], Patrick Martineau[2], Jonathon S. Wright[3], Marta Abalos[4], Petr Šácha[5], Yoshio Kawatani[1], Sean M. Davis[6], Thomas Birner[7,8], and Beatriz M. Monge-Sanz[9,10]

[1]Faculty of Environmental Earth Science, Hokkaido University, Sapporo, 060-0810, Japan
[2]Japan Agency for Marine-Earth Science and Technology, Yokohama, 236-0001, Japan
[3]Department of Earth System Science, Tsinghua University, Beijing, 100084, China
[4]Department of Earth Physics and Astrophysics, Universidad Complutense de Madrid, Madrid, 28040, Spain
[5]Department of Atmospheric Physics, Faculty of Mathematics and Physics, Charles University, Prague, 180 00, Czech Republic
[6]NOAA Chemical Sciences Laboratory, Boulder, CO, 80305, USA
[7]Meteorological Institute, Ludwig-Maximilians-Universität München, Munich, Germany
[8]Institute of Atmospheric Physics, Deutsches Zentrum für Luft- und Raumfahrt, Oberpfaffenhofen, Germany
[9]Atmospheric Oceanic and Planetary Physics, Department of Physics, University of Oxford, Oxford, OX1 3PU, United Kingdom
[10]National Centre for Atmospheric Science, OX1 3PU, Oxford, United Kingdom

*Correspondence to*: Masatomo Fujiwara (fuji@ees.hokudai.ac.jp)

**Abstract.** A 30-year (1980–2010) climatology of the major variables and terms of the transformed Eulerian-mean (TEM) momentum and thermodynamic equations is constructed by using four global atmospheric reanalyses, MERRA-2, JRA-55, ERA-Interim, and CFSR. Both the reanalysis ensemble mean (REM) and the differences of each reanalysis from the REM are investigated in the latitude-pressure domain for December-January-February and for June-July-August. For the REM investigation, two residual vertical velocities (the original one and one evaluated from residual meridional velocity) and two mass streamfunctions (from meridional and vertical velocities) are compared, and longwave (LW) and shortwave (SW) radiative heatings are also shown and discussed. For the TEM equations, the residual terms are also calculated and investigated for their potential usefulness, as the residual term for the momentum equation should include the effects of parameterised processes such gravity waves, while that for the thermodynamic equation should indicate the analysis increment. Inter-reanalysis differences are investigated for the mass streamfunction, LW and SW heatings, the two major terms of the TEM momentum equation (the Coriolis term and the Elliassen-Palm flux divergence term), and the two major terms of the TEM thermodynamic equation (the vertical temperature advection term and the total diabatic heating term). The spread among reanalysis TEM momentum balance terms is around 10 % in Northern-Hemisphere winter and up to 50 % in Southern-Hemisphere winter. The largest uncertainties in the thermodynamic equation (about 50 %) are found in the vertical advection, which does not show a structure consistent with the differences in heatings. The results shown in this paper



provide basic information on the degree of agreement among recent reanalyses in the stratosphere and in the upper troposphere in the TEM framework.

## 1 Introduction

The transformed Eulerian-mean (TEM) set of equations (Andrews et al., 1987; see also Section 2.2 below) is a zonally averaged set of equations of atmospheric motion that describes the zonal mean characteristics of the atmospheric circulation.

The response of the zonal mean flow to eddy momentum and heat fluxes is explicitly shown through the so-called Elliassen-Palm (EP) flux divergence term. The residual-mean meridional circulation ($\overline{v}^*$, $\overline{\omega}^*$ or $\overline{w}^*$) that appears in the TEM equations is known to be a very good approximation of the global mass circulation, also known as the Brewer-Dobson (BD) circulation (Butchart, 2014).

When investigating the residual-mean meridional circulation, EP flux, and other TEM variables and terms in the real atmosphere, researches typically use global meteorological analysis data, or more specifically global atmospheric reanalysis data (e.g. SPARC, 2022 and the references therein) as those variables and terms are not directly observable. However, there are different versions of reanalyses from different reanalysis-producing centres, and different reanalyses may show substantially different results for the same diagnostics due to different methodological details of the reanalysis systems

(SPARC, 2022). SPARC (2022) provided comparisons of some key TEM variables and terms among different reanalyses at climatological/seasonal time scales: tropical upwelling at 70 hPa and EP flux divergence for the 100–70 hPa and 50–3 hPa regions (in its Chapter 5); the residual-mean meridional circulation ($\overline{v}^*$, $\overline{w}^*$) as well as temperature and zonal wind up to the 0.1 hPa level (Chapter 11; see also Chapter 3 for more detailed analysis for temperature and horizontal winds up to 1 hPa); among others. Also, diabatic heatings in the tropical upper troposphere and lower stratosphere were evaluated in Chapters 5

and 8, and ozone data products in the whole stratosphere in Chapter 4.

In this paper, we evaluate all the major variables and terms of the TEM momentum and thermodynamic equations from four reanalysis data sets at climatological time scales, focusing on their latitude-pressure distribution in the December-January-February (DJF) and June-July-August (JJA) seasons. The period is from December 1980 to February 2010 for DJF, and from

1981 to 2010 for JJA. Results for the two equinoctial seasons, March-April-May (MAM) and September-October-November (SON), both for 1981–2010, are shown in the Supplement. The distributions of longwave (LW) and shortwave (SW) radiative heatings in DJF and JJA are also investigated. The monthly imbalance of the TEM momentum equation is also expressed as a residual term which is mainly due to the sub-grid-scale processes such as (parameterised) gravity wave drag (Sato and Hirano, 2019). The monthly imbalance of the TEM thermodynamic equation is expressed as a residual term which

is mainly due to the so-called analysis increment, calculated as the difference between the analysis state and the first guess





(forecast) background state (see e.g. Sections 2.3.1 and 12.1.3 of SPARC, 2022). Note that part of these residuals also arises from the use of interpolated pressure-level data rather than model-level and model-grid data at all model time steps.

The four reanalyses analysed in this paper are: the Modern Era Retrospective-Analysis for Research and Applications,
Version 2 (MERRA-2; Gelaro et al., 2017), the Japanese 55-year Reanalysis (JRA-55; Kobayashi et al., 2015), the European Centre for Medium-Range Weather Forecasts (ECMWF) interim reanalysis (ERA-Interim; Dee et al., 2011), and the Climate Forecast System Reanalysis (CFSR; Saha et al., 2010). Chapter 2 of SPARC (2022) also summarizes the information on key components of all four of these reanalysis systems including the forecast model, assimilation scheme, and observational data assimilated. More recent reanalyses, i.e. ERA5 and JRA-3Q, will be evaluated in a separate paper. For these four reanalyses,
Chapter 5 (Section 5.5.1.1, including Figures 5.4–5.7) of SPARC (2022) showed the following points regarding the climatology of ($\bar{v}^*$, $\bar{w}^*$) and EP flux divergence:

- The annual cycle of the SH part of tropical upwelling is weakest for CFSR and strongest for JRA-55, with ERA-Interim and MERRA-2 in-between.
- The annual cycle of the NH part of tropical upwelling is much smaller than that of the SH counterpart and very
different among reanalyses, with inter-reanalysis spread greater than the seasonal variations.
- The annual cycle of the EP flux divergence averaged for the shallow branch (100–70 hPa) and deep branch (50–3 hPa) (Birner and Bönisch, 2011), separately for the entire NH and SH, shows relatively small inter-reanalysis differences.

Furthermore, Chapter 11 of SPARC (2022) investigated climatological ($\bar{v}^*$, $\bar{w}^*$) for MERRA-2, JRA-55, and ERA-Interim as well as older reanalyses MERRA, JRA-25, and ERA40, concluding that the newer reanalyses MERRA-2, JRA-55, and
ERA-Interim should be used to study transport by the residual circulation. In the current paper, we show some additional results for these variables that complement those in Chapters 5 and 11 of SPARC (2022).

The remainder of this paper is organized as follows. Section 2 introduces the reanalysis data sets analysed in this paper and describes the diagnostics evaluated, namely, the variables and terms of the TEM momentum and thermodynamic equations.
In Section 3, we first discuss DJF results, and then JJA results; for each season, we first discuss the results for the reanalysis ensemble mean (REM), and then the differences of each reanalysis from the REM. Section 4 summarises the findings.

## 2 Data and Method

### 2.1 Reanalysis data

The global atmospheric reanalysis data sets analysed in this paper are, as described in the previous section, MERRA-2, JRA-55, ERA-Interim, and CFSR. The zonal mean diagnostics (see Section 2.2) calculated from these reanalysis data sets are provided by Martineau (2022; M22 hereafter) and Wright (2017; W17 hereafter). Martineau et al. (2018) have provided





detailed information on these zonal mean data sets. The M22 data set, referred to as the Reanalysis Intercomparison Dataset (RID), is an updated and enhanced version of that by Martineau (2017; M17 hereafter) as described by Martineau et al. (2018). The M22 data set newly includes time derivatives of zonal wind and potential temperature and the terms of the TEM thermodynamic equation, except for the diabatic heatings that W17 provides. Note that both W17 and M22 are based on pressure-level data provided by each reanalysis centre. One important difference between M17 data and M22 data is that the former strictly uses a three-point stencil to evaluate all derivatives and thus has missing data regions near/at the poles and in the lower and upper boundary regions, while the latter provides values of derivatives also in such regions; these values are, however, sometimes unrealistic. Therefore, in this paper, we use M22 data but mask data so that the regions with missing data are with the same as those in the M17 dataset. Also, in this paper we analyse monthly means of the common grid data, with the same latitudinal grids (2.5° resolution) and pressure levels for all reanalyses, for both M22 and W17 data sets. Note that W17 mistakenly provided JRA-55 common grid data on a finer latitudinal and vertical grid; we have subset these data to the common grid points for use in this paper. Tables 1 and 2 of Martineau et al. (2018) show the original horizontal grid resolution and all pressure levels for the original grid data, along with the pressure levels corresponding to the common grid.

The W17 data set provides total diabatic heating, diabatic heating due to LW radiation, and that due to SW radiation separately (see Martineau et al., 2018, Section 3.6, for detailed explanation).

We also analyse monthly and zonal mean ozone data from all four reanalyses prepared by Davis (2020) (see also Chapter 4 of SPARC, 2022) in conjunction with SW radiative heating data. These analysed ozone distributions are provided to the forecast model for use in radiation calculations for MERRA-2, JRA-55, and CFSR; however, for ERA-Interim, climatological ozone distributions are used instead (see e.g., Chapters 2 and 4 of SPARC, 2022). Therefore, the analysis results for ERA-Interim ozone in this paper are for reference purposes only.

## 2.2 Calculation of the TEM variables and terms

In the following we primarily use pressure coordinates because we use pressure-level data products in this paper, although the vertical axes for all following figures use the logarithm of pressure. Symbols used below follow the definitions of Martineau et al. (2018) except for those explicitly defined here. Note again that we use the common grid data for all reanalyses with a top level at 1 hPa; see Table 2 and Figure 1 of Martineau et al. (2018) for actual pressure levels considered and the calculation of diagnostics including derivatives, respectively. The residual mean meridional circulation ($\bar{v}^*, \bar{\omega}^*$) in pressure coordinates is defined as

$$\bar{v}^* = \bar{v} - \frac{\partial}{\partial p}\left(\frac{\overline{v'\theta'}}{\frac{\partial \bar{\theta}}{\partial p}}\right) \tag{1}$$



$$\bar{\omega}^* = \bar{\omega} + \frac{1}{a \cos \phi} \frac{\partial}{\partial \phi} \left( \frac{\overline{v'\theta'} \cos \phi}{\frac{\partial \bar{\theta}}{\partial p}} \right) \tag{2}$$

(Martineau et al., 2018). The M22 RID data set includes $\bar{v}^*$ and $\bar{\omega}^*$ (with the latter in the units Pa s$^{-1}$). While $\bar{\omega}^*$ is the vertical wind in pressure coordinates, it is often useful to see the values of vertical wind in log-pressure coordinates, $\bar{w}^*$ in the units m s$^{-1}$. The conversion from $\bar{\omega}^*$ to $\bar{w}^*$ is

$$\bar{w}^* = -\frac{H}{p} \bar{\omega}^* \tag{3}$$

where $H$ is a mean scale height usually set to be 7 km in middle atmosphere studies (Andrews et al., 1987).


The primitive-equation version of the TEM momentum equation is written as

$$\frac{\partial \bar{u}}{\partial t} = f\bar{v}^* - \bar{v}^* \frac{1}{a \cos \phi} \frac{\partial (\bar{u} \cos \phi)}{\partial \phi} - \bar{\omega}^* \frac{\partial \bar{u}}{\partial p} + \frac{1}{a \cos \phi} \nabla \cdot F + \overline{\varepsilon_u} \tag{4}$$

where $F$ is the Eliassen-Palm (EP) flux from waves resolved by the reanalysis, i.e. including Rossby and synoptic-scale waves but excluding the majority of the gravity wave spectrum (see Eqs. (7) and (8) of Martineau et al., 2018 for the

definition of EP flux and its divergence for the primitive-equation version). Since a common 2.5° resolution grid is used, the contributions of smaller-scale waves captured on the finer grids used by some reanalyses are excluded. $\overline{\varepsilon_u}$ is the residual term which includes the effects of parameterized processes such as gravity waves (Sato and Hirano, 2019) and diffusion, effects arising from analysis increments, effects associated with using previously interpolated pressure-level data, and errors in the numerical methods (i.e. to evaluate all derivatives). The M22 RID data set includes all the terms of this equation

except for $\overline{\varepsilon_u}$ which is calculated in this paper as the residual from all other terms in Eq. (4) based on monthly means.

The TEM thermodynamic equation is written as

$$\frac{\partial \bar{\theta}}{\partial t} = -\bar{v}^* \frac{1}{a} \frac{\partial \bar{\theta}}{\partial \phi} - \bar{\omega}^* \frac{\partial \bar{\theta}}{\partial p} - \frac{\partial}{\partial p} \left( \frac{\overline{v'\theta'} \frac{\partial \bar{\theta}}{\partial \phi}}{a \frac{\partial \bar{\theta}}{\partial p}} + \overline{\omega'\theta'} \right) + Q_{total} + \bar{\varepsilon}_\theta \tag{5}$$

where $Q_{total}$ is the sum of all diabatic heatings provided by each reanalysis product (see Section 2.1) and $\bar{\varepsilon}_\theta$ is the residual

term which includes the effects of analysis increments, effects associated with using pressure-level data, and errors in the numerical methods. The summation of the first three terms on the right-hand side of this equation is mathematically equivalent to the summation of the second to fourth terms on the left-hand side of Equation 12 in Martineau et al. (2018), which is the Eulerian mean, not TEM. The M22 RID data set includes all terms of Eq. (5) except for $Q_{total}$ and $\bar{\varepsilon}_\theta$. For $Q_{total}$, we use W17 data set (see Martineau et al., 2018). $\bar{\varepsilon}_\theta$ is calculated in this paper as the residual from all other terms of

Eq. (5) based on monthly means. It is also noted that $\bar{\varepsilon}_\theta$ is mathematically the same as $\bar{\chi}$ in Equation 12 of Martineau et al. (2018), although they are numerically different (see the Supplement Folder 1) owing to numerical differences between the



summation of the first three terms on the right-hand side of Eq. (5) and the summation of the second to fourth terms on the left-hand side of Equation 12 in Martineau et al. (2018).

Considering the TEM continuity equation,

$$\frac{1}{a \cos\phi}\frac{\partial}{\partial\phi}(\bar{v}^* \cos\phi) + \frac{\partial\bar{\omega}^*}{\partial p} = 0 \tag{6}$$

we can define a streamfunction $\Psi_p^*$ for pressure coordinates (in the units, Pa m s$^{-1}$) as

$$\bar{v}^* = +\frac{1}{\cos\phi}\frac{\partial\Psi_p^*}{\partial p} \tag{7}$$

$$\bar{\omega}^* = -\frac{1}{a \cos\phi}\frac{\partial\Psi_p^*}{\partial\phi} \tag{8}$$

Therefore, with appropriate boundary conditions, we can calculate $\Psi_p^*$ from one of the followings:

$$\Psi_p^* = +\cos\phi \int_{TOA}^{p} \bar{v}^* \, dp' \tag{9}$$

$$\Psi_p^* = -a \int_{SP}^{\phi} \bar{\omega}^* \, d\phi' \tag{10}$$

$$\Psi_p^* = -a \int_{NP}^{\phi} \bar{\omega}^* \, d\phi' \tag{11}$$

where $TOA$ stands for the nominal top of atmosphere, $SP$ for the South Pole, and $NP$ for the North Pole (note that $d\phi'$ is
negative in Eq. (11)). $\Psi_p^*$ calculated from $\bar{v}^*$ is often used in middle atmosphere studies (e.g., Abalos et al., 2015) because $\bar{v}^*$ data may be more reliable than $\bar{\omega}^*$ in reanalysis data (as meridional wind observations are assimilated, while vertical winds are not). On the other hand, values of $\Psi_p^*$ calculated from $\bar{v}^*$ are rather sensitive to the treatment of upper boundary conditions (i.e., $TOA$ in the integral); in some cases they are sensitive even down to the lower stratosphere depending on the height of the top data level. Thus, some works (e.g., Sato and Hirano, 2019) use $\Psi_p^*$ calculated from $\bar{\omega}^*$. In this paper, we
calculate both streamfunctions and compare the two. When calculating $\Psi_p^*$ calculated from $\bar{v}^*$, we follow Chapter 5 of SPARC (2022, Section 5.2.1) for the treatment of the upper boundary (i.e., $TOA$ in the integral). In short, we create monthly $\bar{v}^*$data at the top two levels (1 hPa and 2 hPa for the common grid data set), where they are missing in M17, by extrapolation and with some assumptions. We then set the top boundary conditions at 0 hPa and in the 0–1 hPa layer so that the average $\bar{v}^*$ for the 0–1 hPa layer is half the $\bar{v}^*$at 1 hPa, which corresponds to setting $\bar{v}^* = 0$ at 0 hPa. For $\Psi_p^*$ calculated from $\bar{\omega}^*$, we use
Eq. (10) for the Southern Hemisphere (SH) and Eq. (11) for the Northern Hemisphere (NH) and set values at the equator to the average of values from Eqs. (10) and (11).



In Section 3, as for many previous studies (e.g., Abalos et al., 2015; Sato and Hirano, 2019; Chapter 5 of SPARC, 2022), the streamfunction in log-pressure coordinates, or the mass streamfunction ($\Psi^*$), in the units kg m$^{-1}$ s$^{-1}$ is shown. The conversion to the mass streamfunction is

$$\Psi^* = \frac{H}{RT_s}\Psi_p^* = \frac{1}{g_0}\Psi_p^* \qquad (12)$$

where $R$ is the gas constant for dry air, $T_s$ is a constant reference temperature set as 240 K, and $g_0$ is the global average gravitational constant at mean sea level (Andrews et al., 1987, Sections 1.1.1 and 3.1.1). Hereafter, the mass streamfunction calculated from $\bar{v}^*$ is referred to as $\Psi^*_{\bar{v}^*}$, and that calculated from $\bar{\omega}^*$ as $\Psi^*_{\bar{\omega}^*}$.

Finally, we also calculate $\bar{\omega}^*$ and $\bar{w}^*$ from $\Psi^*_{\bar{v}^*}$ through Eqs. (8) and (3) ($\bar{\omega}^*_{\bar{v}^*}$ and $\bar{w}^*_{\bar{v}^*}$, respectively) and compare them with the original $\bar{\omega}^*$ and $\bar{w}^*$ in Section 3.

**2.3 Climatological tropopause location**

The climatological latitudinal distribution of tropopause pressure is shown in the figures in Section 3. The tropopause is defined here as the lowermost location above 5 km altitude where the temperature gradient with respect to log-pressure height ($z = -H\ln(p/p_s)$, with $p_s = 10^5$ Pa) becomes $-2$ K km$^{-1}$, using linear interpolation to find the exact point. The same definition is used at both tropical and extratropical latitudes. Note that we used the 30-year (1981–2010) climatological mean temperature distributions from  monthly-averaged common grid reanalysis data to determine the climatological tropopause locations. Therefore, the tropopause as shown in the following figures is for illustrative purposes only.



## 3 Results and Discussion

### 3.1 DJF

### 3.1.1 REM for DJF

Figure 1 shows the REM climatological latitude-pressure distributions of the TEM variables for DJF. Values in the lower troposphere are often missing because MERRA-2 does not provide pressure-level data below the Earth surface and because 210 zonal means were not calculated in M22 for latitude bands with one or more missing data points in longitude. During DJF, the tropical tropopause is colder than all other seasons, the NH polar lower stratosphere is colder than the SH polar lower stratosphere, and the SH upper stratosphere is warmer than the NH upper stratosphere. The distributions of temperature and zonal wind agree quite well with the thermal wind balance in the zonal mean (not shown directly). The residual mean meridional circulation (i.e. the advective part of the stratospheric BD circulation) shows the following characteristics: (1) 215 upwelling in the tropics (with two maxima, one around 10°N and the other around 30°S, and a minimum in the equatorial lower stratosphere; note that the closed contours around 70–30 hPa at the equator in $\overline{w}^*$ show a minimum; see also Chapter 5 of SPARC (2022), their Figures 5.2 and 5.5); (2) poleward flow in the stratosphere, i.e. northward flow in the NH and southward flow in the SH; and (3) downwelling in the extratropics. The NH northward flow is much stronger than the SH southward flow during DJF. The $\overline{v}^*$ distribution also clearly shows the shallow branches of the BD circulation in the 220 midlatitude lower stratosphere (200–100 hPa in NH and 200–50 hPa in SH). Within these distributions, we also see the upper tropospheric branch of the Hadley cells in the tropics, with the tropical-to-NH (clockwise) cell being stronger during DJF (see e.g. Schneider and Bordoni, 2008). Equatorward flow along the midlatitude tropopause in both hemispheres is evident in all four reanalyses (see Supplement Folder 2), and is associated with EP flux divergence due to resolved waves there (see Figure 4) as discussed by Birner et al. (2013).


Figure 1 also compares $\overline{w}^*$ and $\overline{w}^*_{\overline{v}^*}$, the latter of which is estimated from $\overline{v}^*$ through the streamfunction calculation (i.e. through the continuity equation). The two vertical velocity fields show reasonable agreement in the troposphere and in the lower stratosphere up to 10 hPa, but differences even with this roughly logarithmic contouring are evident in the upper stratosphere. In general, reanalysis meridional wind products are strongly constrained by observations through data 230 assimilation. By contrast, vertical velocities in reanalysis products are highly dependent on the specific implementation of data assimilation. For example, in early reanalyses using 3-dimensional variational (3D-Var) assimilation, vertical winds are primarily determined by the underlying forecast model (e.g. Section 6 of Kalnay et al., 1996). In more recent reanalysis systems using 4D-Var assimilation techniques, vertical velocities are influenced by observational data indirectly through





data assimilation constraints on horizontal winds. Because vertical velocities are small and computed indirectly from
horizontal divergence, even small assimilation increments in horizontal winds can have large influences on vertical velocities
(Uma et al., 2021). These effects can produce substantial noise in reanalysis estimates of vertical velocity (Wohltmann and
Rex, 2008; Hoffmann et al., 2019). Monge-Sanz et al. (2007, 2012) showed how advances in the assimilation schemes
resulted in more realistic vertical wind fields, and also that improvements were still needed. Therefore, estimates of $\overline{\omega}^*/\overline{w}^*$
from $\overline{v}^*$ may still be more reliable for studies of particular atmospheric processes in particular regions. It should be noted
that estimation through the streamfunction has its own issues, as the streamfunction from Eq. (9) is sensitive to conditions
applied for the "$TOA$" (including the choice of the data top) even down to the lower stratosphere. Therefore, looking at both
estimates of residual vertical velocity and trusting only the common features may be a good approach. Note also that in this
paper we use the common grid data set for which the top is located at 1 hPa for the purpose of comparisons of different
reanalyses.The use of original-grid data (or model-level data) with higher tops would improve estimates of $\overline{\omega}^*/\overline{w}^*$ from $\overline{v}^*$.)


Figure 1 also shows and compares the two streamfunctions $\Psi^*_{\overline{v}^*}$ and $\Psi^*_{\overline{\omega}^*}$ (see Section 2.2 for the details). During DJF, the
NH cells for both the BD circulation and the Hadley circulation are more pronounced than their SH counterparts. This is in
overall agreement with the results from satellite Michelson Interferometer for Passive Atmospheric Sounding (MIPAS)
observations (von Clarmann et al., 2021), with a cautionary note that their climatology is for the period 2002–2012. We also
notice quantitative differences among the four reanalyses with a roughly logarithmic contouring in Fig. 1 not only in the
upper stratosphere (above the 10 hPa level) but also in the lower stratosphere. These differences arise due to the reasons
discussed in the previous paragraph.



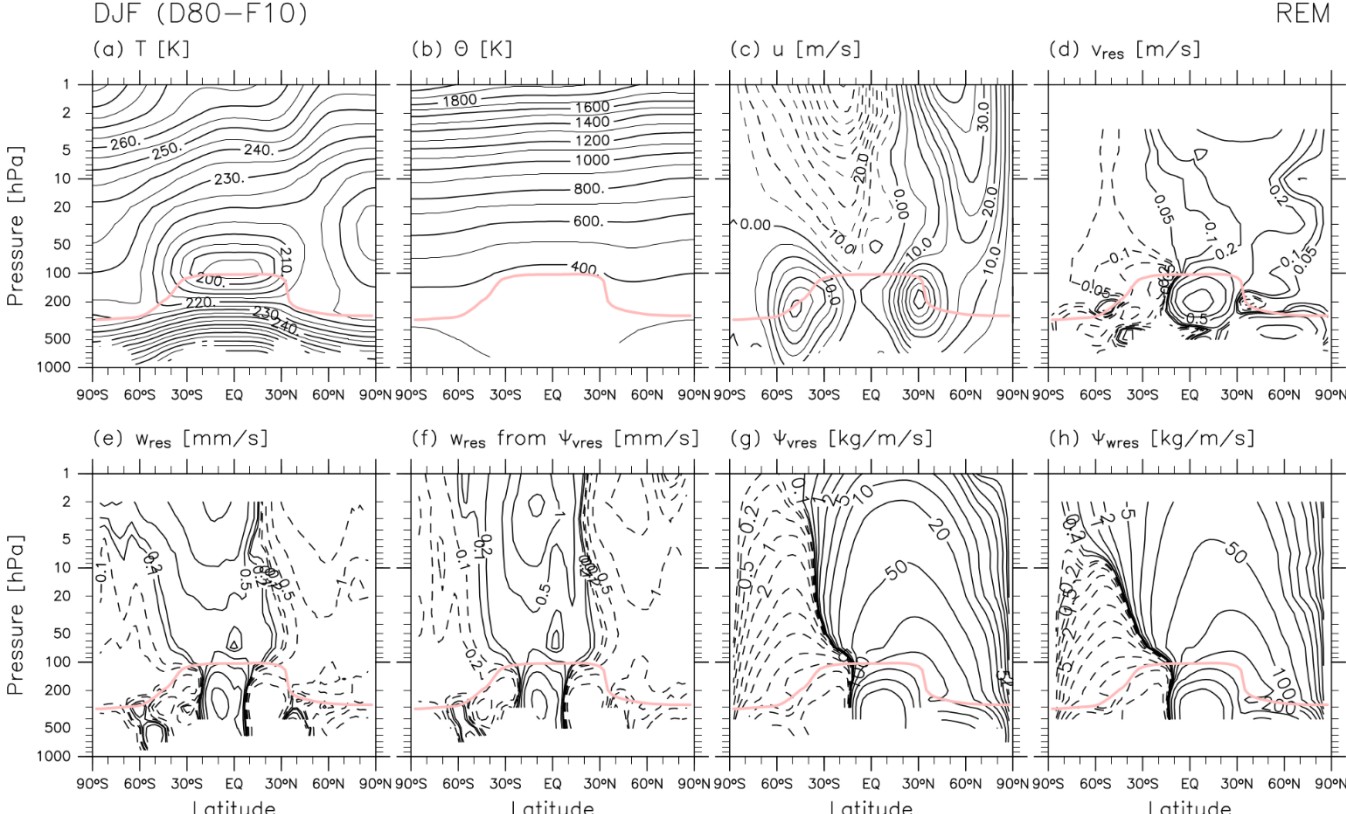

**Figure 1: Latitude-pressure distributions of the REM for 30-year DJF means (December 1980–February 1981 to December 2009– February 2010) of zonal-mean (a) temperature (contour interval: 5 K), (b) potential temperature (contour interval: 100 K), (c) zonal wind (contour interval: 5 m s$^{-1}$, with dotted for negative/westward), (d) $\overline{v}^*$ (contours: ±0.05, ±0.1, ±0.2, ±0.5, ±1, . . . m s$^{-1}$, with dotted for negative/southward), (e) $\overline{w}^*$ (contours: ±0.1, ±0.2, ±0.5, ±1, ±2, . . . mm s$^{-1}$, with dotted for negative/downward), (f) $\overline{w}^*_{\overline{v}^*}$ (contours and dotted: same as for Fig. 1(e)), (g) $\Psi^*_{\overline{v}^*}$ (contours: ±0.1, ±0.2, ±0.5, ±1, ±2, . . . kg m$^{-1}$ s$^{-1}$, with dotted for negative/anticlockwise), and (h) $\Psi^*_{\overline{\omega}^*}$ (contours and dotted: same as for Fig. 1(g)). See Section 2.2 for the details of the two different vertical wind estimates and the two different mass streamfunctions. The pink curve in all panels shows the location of the DJF-mean climatological tropopause based on the REM.**



Figure 2 shows REM climatological distributions of diabatic heatings for DJF, with particular attention to the radiative heatings. Note that all heatings shown in Figure 2 are with respect to temperature tendency, not potential temperature tendency, to facilitate comparison with previous literature. Andrews et al. (1987, Chapter 2) discuss radiative heatings in the stratosphere and lower mesosphere based on results from e.g. Kiehl and Solomon (1986) who used radiative transfer models and satellite observations of temperature and ozone. More recent assessments of middle-atmosphere radiative heatings include those by Gettelman et al. (2004), Fueglistaler et al. (2009), SPARC (2010, Chapter 3), Ming et al. (2016), and Tao et al. (2019; their Fig. 3). For LW heating, major contributions in the stratosphere include cooling to space by $CO_2$ (roughly three-fourths) and $O_3$ (roughly a fourth ), with that by $H_2O$ having a non-negligible contribution (Andrews et al., 1987, their Fig. 2.1). Weak positive LW heatings around the tropical tropopause region are due to absorption of fluxes from below by $O_3$. Negative LW heatings in the troposphere are mainly attributable to $H_2O$. For SW heating, absorption by $O_3$ is the major component in the stratosphere, together with the latitudinal and seasonal distribution of solar insolation at the $TOA$, which is much greater in the SH than in the NH during DJF (see e.g. Liou, 2002). The REM ozone distribution for DJF is also shown in Figure 2 for reference (see the caveat in the last paragraph of Section 2.1). Other components of diabatic heating include convective heating and large-scale condensation heating, primarily in the troposphere, heating by turbulent mixing in regions of shear-flow instability, and heating due to parameterized gravity wave drag. In the stratosphere, the distribution of the total diabatic heating is almost entirely determined by the balance between LW cooling and SW heating (Fig. 2d-e). Although the total diabatic heating is nearly zero in the global mean (Fig. 2e), during DJF it is comprised of heating in the SH stratosphere and cooling in the NH stratosphere.



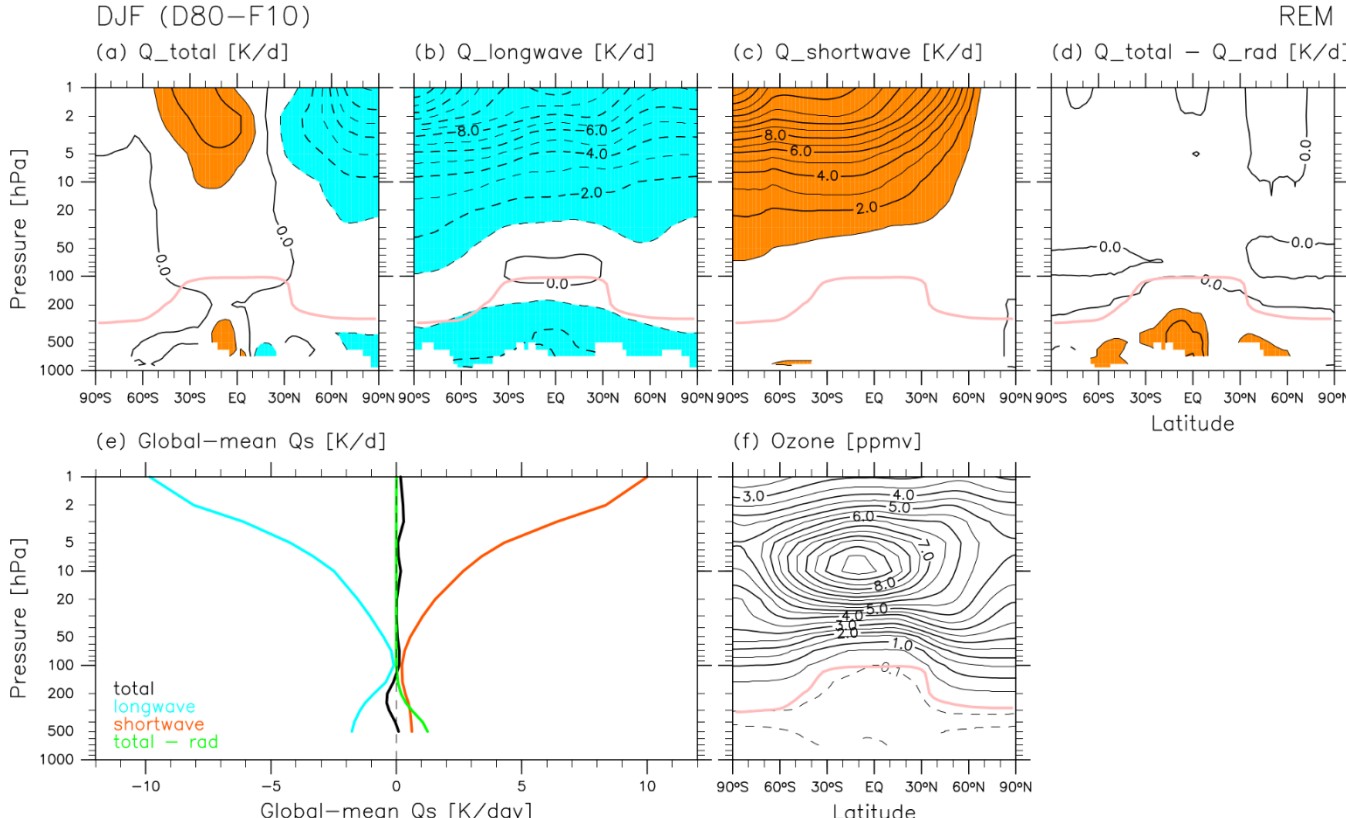

**Figure 2: Latitude-pressure distributions of the REM for 30-year DJF means (December 1980–February 1981 to December 2009–February 2010) of zonal-mean (a) total diabatic heating (in terms of temperature, not potential temperature; the same for Figs. 2(b)–2(e)), (b) longwave radiative heating, (c) shortwave radiative heating, and (d) diabatic heating due to processes other than radiative transfer. The contour interval in Figs. 2(a)–2(d) is 1 K day⁻¹, with dotted contours for negative values; regions with values greater than +1 K day⁻¹ are coloured in orange, while those with values smaller than −1 K day⁻¹ are coloured in light blue. The pink curve in all panels shows the location of the REM DJF-mean climatological tropopause. (e) Vertical distribution of global-mean diabatic heating (black: total; light blue: longwave radiative; orange: shortwave radiative; light green: other than radiative). (f) As for Fig. 2(a) but for ozone mixing ratio (contour interval is 0.5 parts per million by volume (ppmv), with the 0.1 and 0.05 ppmv contours shown as dotted lines).**






Figure 3 shows the REM climatological distributions of all terms in the TEM momentum equation, Eq. (4), in units of m s$^{-1}$ day$^{-1}$, for DJF. The sign of each term is defined as in Eq. (4). The major terms in the stratosphere at monthly time scales are the Coriolis term and the EP flux divergence term (the latter due to resolved waves), with strong signals extending much higher in the NH than in the SH during this season. These results illustrate the main mechanisms driving the BD circulation,

namely, that EP flux convergence arising mainly from the dissipation of upward propagating Rossby waves in the extratropical stratosphere and synoptic-scale waves in the subtropical lower stratosphere results in poleward flow (Section 4 of Butchart, 2014). During DJF, the existence of a polar night jet in the NH (Fig. 1(c)) enables Rossby waves to propagate higher in the NH stratosphere, producing greater EP flux convergence and driving stronger poleward flow in the NH. Along the midlatitude tropopause in both hemispheres, signals in the EP flux divergence due to resolved waves correspond to

equatorward flow (see both Figures 1 (d) and 4 (b); Birner et al., 2013). The momentum balance in the troposphere is more complicated in this TEM framework, with additional contributions from the meridional and vertical advection terms. As noted in Section 2.2, the residual term $\overline{\varepsilon_u}$ includes the effects of parameterized processes such as gravity waves and diffusion, errors in the numerical methods, and adjustments arising from analysis increments. The main contribution to $\overline{\varepsilon_u}$ in the stratosphere comes from the forcing due to dissipating gravity waves (Sato and Hirano, 2019), while contributions from

diffusion and cloud processes may also be important in the troposphere. Negative signals in $\overline{\varepsilon_u}$ in the midlatitude lower stratosphere may result in part from unresolved forcing due to gravity waves generated by the subtropical jets (e.g. Kawatani et al., 2004; Plougonven and Snyder, 2007) and the orography (e.g. Kuchar et al., 2020). See also Podglajen et al., 2020 for a comparison of reanalyses with long-duration, quasi-Lagrangian balloon observations in the equatorial and Antarctic lower stratosphere with respect to gravity wave spectra. We also find negative signals in the NH high-latitude upper stratosphere,

which may result in part from gravity waves generated by the winter polar night jet and the orography.



**Figure 3: Latitude-pressure distributions of the REM for 30-year DJF means (December 1980–February 1981 to December 2009–February 2010) of each term in the TEM momentum equation, Eq. (4): (a) zonal wind tendency term, (b) Coriolis term, (c) meridional advection term, (d) vertical advection term, (e) EP flux divergence term, and (f) the residual term $\overline{\varepsilon_u}$. The sign of each term is defined as that shown in Eq. (4). Contours are located at $\pm 0.1$, $\pm 0.2$, $\pm 0.5$, $\pm 1$, $\pm 2$, . . . m s$^{-1}$ day$^{-1}$ with dotted contours for negative values in all panels; orange shading indicates values greater than 0.5 m s$^{-1}$ day$^{-1}$, while light blue shading indicates values smaller than −0.5 m s$^{-1}$ day$^{-1}$. The pink curve in all panels shows the location of the REM DJF-mean climatological tropopause.**





Figure 4 shows REM climatological distributions of all terms in the TEM thermodynamic equation, Eq. (5), in units of K day$^{-1}$ during DJF. The sign of each term is defined as in Eq. (5). The major terms in the stratosphere at monthly time scales are the vertical advection term and the total diabatic heating term (essentially radiative heating, as shown in Figure 2), but other terms show noticeable contributions at higher latitudes in the middle to upper stratosphere. Most notably, values of the residual term $\overline{\varepsilon_\theta}$ are on the same order of magnitude as those for the two major terms in the NH stratosphere during DJF. As noted in Sections 1 and 2.2, the main component of $\overline{\varepsilon_\theta}$ is the analysis increment, which is the difference between the analysis state and the first guess (forecast) background state. Figure 4(f) indicates that there are large differences between the observations and the forecast models in the NH mid-to-upper stratosphere during DJF.



**Figure 4: Latitude-pressure distributions of the REM for 30-year DJF means (December 1980–February 1981 to December 2009– February 2010) of each term in the TEM thermodynamic equation, Eq. (5): (a) potential temperature tendency term, (b) meridional advection term, (c) vertical advection term, (d) wave flux term (the third term of right-hand side of Eq. (5)), (e) total diabatic heating term, and (f) the residual term $\overline{\varepsilon_\theta}$. The sign of each term is defined as in Eq. (5). Contours are located at ±0.1, ±0.2, ±0.5, ±1, ±2, . . . K day$^{-1}$ with dotted contours for negative values in all panels; orange shading indicates values greater than 0.5 K**





day$^{-1}$, while light blue shading indicates values smaller than $-0.5$ K day$^{-1}$. The pink curve in all panels shows the location of the REM DJF-mean climatological tropopause.



### 3.1.2 Differences of each reanalysis from REM for DJF

The variables and terms discussed in this section include the mass streamfunction of the residual mean meridional circulation calculated from $\bar{v}^*$ ($\Psi^*_{\bar{v}^*}$), LW and SW radiative heatings, the two major terms of the TEM momentum equation, and the two major terms of the TEM thermodynamic equation. Differences with respect to the REM for each reanalysis are shown in the following figures, as along with inter-reanalysis spreads presented as standard deviation (SD) and relative SD (i.e. SD divided by the absolute value of REM). See the Supplement Folder 3 for other major TEM variables and terms including temperature and zonal wind: Note that for temperature, the differences among different reanalyses become greater at higher altitudes (because of weaker observational constraints), and for zonal wind, the differences are largest in the tropics (because of a weaker thermal-wind constraint) and in the low-to-midlatitude upper stratosphere, as also shown in Chapters 3 and 11 of SPARC (2022). Figure 5 shows differences for the mass streamfunction $\Psi^*_{\bar{v}^*}$ during DJF. The differences change sign across latitudes, suggesting differences in the structure of the residual-mean meridional circulation among different reanalyses (e.g. the separation location between the shallow and deep branches). In the lower stratosphere below the 10 hPa level, the main, NH cell of the BD circulation (Fig. 1(g)) is generally stronger for JRA-55 and weaker for MERRA-2. This tendency can be seen in the different distributions of $\bar{v}^*$ (Supplement Folder 3). Inter-reanalysis standard deviations relative to the REM (Fig. 5(f)) indicate differences of 2–10 % among these reanalyses in the main, NH cell of the BD circulation. Note that these fractional differences can be quite large in regions where the REM is close to zero; thus we must always refer back to the REM distribution to identify the important regions. The features for $\Psi^*_{\bar{v}^*}$ described above are generally in good agreement with those for $\Psi^*_{\bar{\omega}^*}$ (Supplement Folder 3). Differences in the individual components of the residual circulation ($\bar{v}^*, \bar{w}^*$) can also be found in the Supplement Folder 3. For example, differences in $\bar{w}^*$ during DJF (the Supplement Folder 3) show vertical bands with widths of roughly 20–30 degrees in latitude and are therefore difficult to characterize.

Figure 6 shows differences in LW radiative heating during DJF. The greatest absolute differences are found in the upper stratosphere (fractional differences of 5–10 % generally and >10 % in the winter polar upper stratosphere), with MERRA-2 and JRA-55 being more positive than the REM and ERA-Interim and CFSR being more negative. Strong negative differences in CFSR and strong positive differences in JRA-55 are consistent with the temperature differences in these two reanalyses; that is, warmer in CFSR and colder in JRA-55 (Supplement; Chapter 3 of SPARC, 2022). In the middle atmosphere, the cooling-to-space (or Newtonian cooling) approximation works well where the LW radiative cooling depends on local temperature (e.g., Liou, 2002, Section 4.5.2). Thus, the LW heating differences in the upper stratosphere shown in Fig. 6 may be largely determined by differences in temperature. By contrast, LW heating differences in the troposphere and around the tropopause are probably related mainly to differences in the distribution of clouds (Fueglistaler and Fu, 2006; Wright et al., 2020; Chapter 8 of SPARC, 2022). Large fractional differences (10–50 % and even largeer in some regions) are found around the tropopause globally and in the tropical-to-subtropical lower stratosphere where heating due to $O_3$ absorption of upwelling LW radiation fluxes from the troposphere is also important (Fig. 5(f)). Figure 7 shows differences in




SW radiative heating during DJF. The greatest absolute differences are found in the sunlit region of the upper stratosphere
(fractional differences of 5–10 %). The strong negative differences in JRA-55 are consistent with negative differences in

380 ozone concentration in this reanalysis (Supplement; Chapter 4 of SPARC, 2022). By contrast, the strong positive differences
in CFSR in comparison with negative differences in MERRA-2 cannot fully understood from differences in ozone
concentrations between these two reanalyses, implying the existence of other factors (such as details of the radiative transfer
schemes; Table 2.4 of SPARC, 2022). Note that ERA-Interim uses climatological ozone distributions for radiative transfer
calculations. Differences in the tropical upper troposphere, where fractional differences exceed 50 %, may be related to

385 differences in the cloud distribution (Wright et al., 2020; Chapter 8 of SPARC, 2022). Large fractional differences (10–
50 %) are also evident around the extratropical tropopause.





**Figure 5: Latitude-pressure distributions of 30-year DJF means (December 1980–February 1981 to December 2009–February 2010) of the $\Psi^*_{\overline{v}^*}$ anomaly with respect to the REM for (a) MERRA-2, (b) JRA-55, (c) ERA-Interim, and (d) CFSR. Contours are located at ±0.1, ±0.2, ±0.5, ±1, ±2, . . . kg m$^{-1}$ s$^{-1}$ with dotted contours for negative values in all panels; orange shading indicates values greater than 1 kg m$^{-1}$ s$^{-1}$, while light blue shading indicates values smaller than −1 kg m$^{-1}$ s$^{-1}$. (e) Inter-reanalysis differences for $\Psi^*_{\overline{v}^*}$ presented as standard deviation (SD; contours at 0.1, 0.2, 0.5, 1, 2, . . . kg m$^{-1}$ s$^{-1}$; light red shading for values greater than 2 kg m$^{-1}$ s$^{-1}$). (f) Inter-reanalysis differences for $\Psi^*_{\overline{v}^*}$ presented as SD divided by the absolute value of the REM in percent (contours are at 1, 2, 5, 10, 20, . . . %, light red shading marks values greater than 10 %, and darkred shading marks values greater than 50 %). The pink curve in all panels shows the location of the DJF-mean climatological tropopause for each reanalysis in panels (a)–(d) and for the REM for in panels (e) and (f).**



**Figure 6: As for Figure 5, but for longwave radiative heating. For Fig. 6(a)–(e), the contour interval is 0.1 K day$^{-1}$ with dotted contours for negative values in all panels. For Fig. 6(a)–(d), orange shading indicates values greater than 0.1 K day$^{-1}$, while light blue shading indicates values smaller than −0.1 K day$^{-1}$. For Fig. 6(e), light red shading marks values greater than 0.1 K day$^{-1}$. For Fig. 6(f), contours are located at 1, 2, 5, 10, 20, . . . %, light red shading marks values greater than 5 %, and dark red shading marks values greater than 10 %.**





**Figure 7: As for Figure 6, but for shortwave radiative heating.**

410



Figures 8 and 9 show differences of each reanalysis relative to the REM during DJF for the two major terms of the TEM momentum equation, i.e. the Coriolis term and the EP flux divergence term. The distribution of differences in the Coriolis term reflects that in $\bar{v}^*$ (Supplement Folder 3). In the midlatitude lower stratosphere below the 10 hPa level, generally positive differences (stronger poleward flows) are found in JRA-55 and negative differences (weaker poleward flow) in MERRA-2. Fig. 8(f) shows that inter-reanalysis fractional differences for $f\bar{v}^*$ are generally less than 10 % in the NH extratropical stratosphere and SH lower stratosphere, where strong poleward flows are found in the REM (Fig. 3(d)). In the winter hemisphere where we expect wave-driven $\bar{v}^*$, Fig. 9 shows that differences in the EP flux divergence term exhibit generally negative differences (more convergence) in JRA-55 and positive differences (less convergence) in MERRA-2. Large differences (both positive and negative) are found in the NH middle-to-upper stratosphere and in the extratropical lower stratosphere in both hemispheres, both regions where the EP flux divergence has significant values in the REM, indicating differences in Rossby and synoptic-scale wave activity across the four reanalyses. Differences in resolved wave activity in the stratosphere can be caused in part by different treatments of unresolved gravity waves in the reanalyses (see differences in the residual term in Supplement Folder 3), which can affect the resolved wave field through a set of dynamical interactions termed as the compensation mechanism by Cohen et al. (2013, 2014). Moreover, Eichinger et al. (2020) have shown that the choice of gravity wave parameterization scheme in a climate model influences the resolved wave field throughout the model domain, often in the opposite sense to compensation. Inter-reanalysis fractional standard deviations are generally less than 10 % in the extratropical stratosphere (Fig. 9(f)). Figures. 3(b) and 9 are complementary to Figure 5.4 in Chapter 5 of SPARC (2022), which shows the seasonal cycles of EP flux divergence averaged for the shallow (100–70 hPa) and deep (50–3 hPa) branches of the BD circulation in the NH and SH separately. Figure 9 indicates that large local inter-reanalysis differences are obscured when they are averaged over the whole hemisphere.

Figures 10 and 11 show differences in each reanalysis relative to the REM during DJF for the two major terms of the TEM thermodynamic equation, i.e. the vertical temperature advection term and the total diabatic heating term. The distribution of differences in the vertical temperature advection term reflects inter-reanalysis differences in $\bar{\omega}^*/\bar{w}^*$ (i.e. vertical bands of positive and negative anomalies, suggesting differences in the structure of the circulation among the reanalyses; Supplement Folder 3) in addition to those in temperature (with greater differences at higher altitudes; Supplement Folder 3). Figure 10(f) shows that fractional inter-reanalysis differences for the vertical temperature advection term are generally less than 50% in locations where the term has absolute values greater than 1 K day$^{-1}$ in the REM (Fig. 4(c)). This result is consistent with the findings of Abalos et al. (2015) who showed ~40 % uncertainty in tropical upwelling magnitude. In the NH midlatitude stratosphere, fractional differences are generally even less than 10 %. Figure 11 shows that differences in the total diabatic heating term show horizontal bands of large positive and negative anomalies in the middle to upper stratosphere and large values in the troposphere; these come from combining inter-reanalysis differences in both LW and SW heatings (Figs. 6 and 7, respectively). Therefore, the distributions of differences in the two major terms of the TEM thermodynamic equation do not correspond well to each other. Figure 11(f) shows that fractional inter-reanalysis differences in the SH net heating region



(see Fig. 4(e)) are generally less than 50 %, while those in the NH net cooling region are generally less than 10 %. These results show that modern global reanalysis systems still need to improve momentum and thermodynamic balance even on the climatological zonal mean scale.



**Figure 8:** As for Figure 5, but for the Coriolis term. For Fig. 8(a)–(e), contours are located at ±0.01, ±0.02, ±0.05, ±0.1, ±0.2, . . . m s$^{-1}$ day$^{-1}$ with dotted contours for negative values in all panels. For Fig. 8(a)–(d), orange shading indicates values greater than 0.05 m s$^{-1}$ day$^{-1}$, while light blue shading indicates values smaller than −0.05 m s$^{-1}$ day$^{-1}$. For Fig. 8(e), light red shading marks values greater than 0.1 m s$^{-1}$ day$^{-1}$. For Fig. 8(f), contours are located at 1, 2, 5, 10, 20, . . . %, light red shading marks values greater than 10 %, and dark red shading marks values greater than 50 %.





**Figure 9: As for Figure 8, but for the EP flux divergence term.**





Figure 10: As for Figure 5, but for the vertical temperature advection term of the TEM thermodynamic equation. For Fig. 10(a)–(e), contours are located at ±0.05, ±0.1, ±0.2, ±0.5, ±1, . . . K day$^{-1}$ with dotted contours for negative values for all the panels. For Fig. 10(a)–(d), orange shading indicates values greater than 0.2 K day$^{-1}$, while light blue shading indicates values smaller than −0.2 K day$^{-1}$. For Fig. 10(e), light red shading indicates values greater than 0.2 K day$^{-1}$. For Fig. 10(f), contours are located at 1, 2, 5, 10, 20, . . . %, light red shading marks values greater than 10 %, and dark red shading marks values greater than 50 %.



**Figure 11: As for Figure 10, but for the total diabatic heating term of the TEM thermodynamic equation.**




## 3.2 JJA

### 3.2.1 REM for JJA

Figure 12 shows the REM climatological latitude-pressure distributions of the TEM variables for JJA. During this season,
the SH polar lower stratosphere becomes quite cold, and the NH upper stratosphere is warmer than the SH upper stratosphere.
As during other seasons, the distributions of temperature and zonal wind agree quite well with the thermal wind balance in
the zonal mean (not shown directly). The BD circulation during this season shows one cell covering the SH and the tropics in
the middle to upper stratosphere (i.e. the upper branch) and two cells in the lower stratosphere and around the tropopause (i.e.
the shallow branches) as observable in both $(\bar{v}^*, \bar{w}^*)$ and mass streamfunction. As during DJF, the tropical upwelling during
this season also has two maxima in the NH and SH subtropics and a minimum in the equatorial lower stratosphere, with the
NH subtropical upwelling being much stronger during JJA (in other words, the summer-side tropical upwelling is stronger).
We also see the upper tropospheric branch of the Hadley cells in the tropics, with the tropical-to-SH (anticlockwise) cell
being stronger during JJA (thus the winter-side cell is always stronger; see also Fig. 1 and e.g. Schneider and Bordoni, 2008).
Northward flow around the SH midlatitude tropopause is evident in all four reanalyses (see Supplement Folder 2), and is
associated with EP flux divergence due to resolved waves there (see Figure 14 and Birner et al., 2013).

Figure 12 also compares $\bar{w}^*$ and $\bar{w}^*_{\bar{v}^*}$ during JJA. Agreement between the two is somewhat weaker compared to that for
DJF (Figure 1), as differences are also more evident in the lower stratosphere. As in Fig. 1, comparison of the two mass
streamfunctions during JJA in Figure 12 indicates quantitative differences not only in the upper stratosphere but also in the
lower stratosphere.



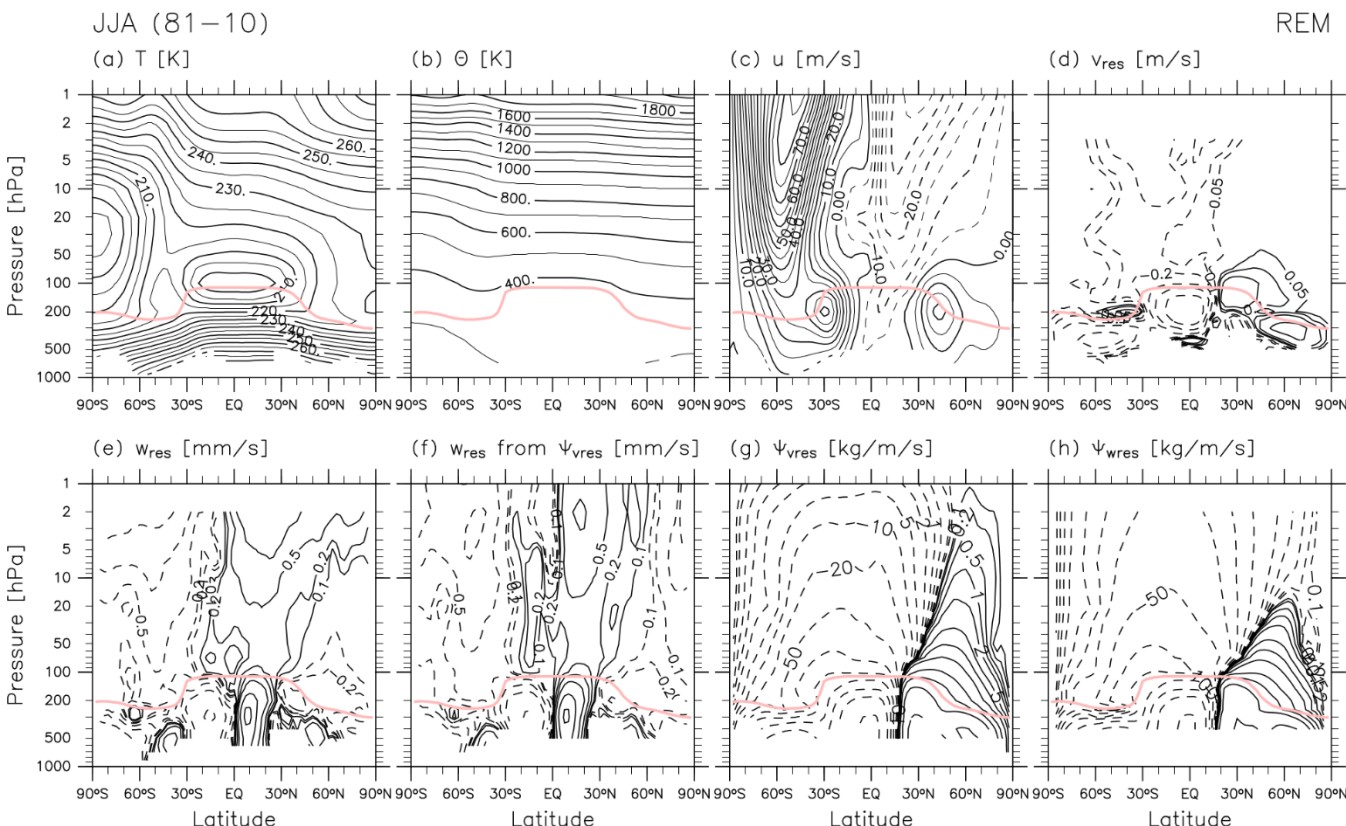

**Figure 12: As for Figure 1, but for the 30-year JJA (1981–2010) mean.**




Figure 13 shows REM climatological distributions of diabatic heatings for JJA. Roughly, in the stratosphere, the distributions of the total and radiative heating are equatorially in mirror image with those for DJF (Figure 2) because of the distribution of solar insolation at the $TOA$ (e.g. Liou, 2002). In the stratosphere, the total (and net radiative) diabatic heating is positive in the tropics and at the mid-latitudes and strongly negative at SH high latitudes at high altitudes.


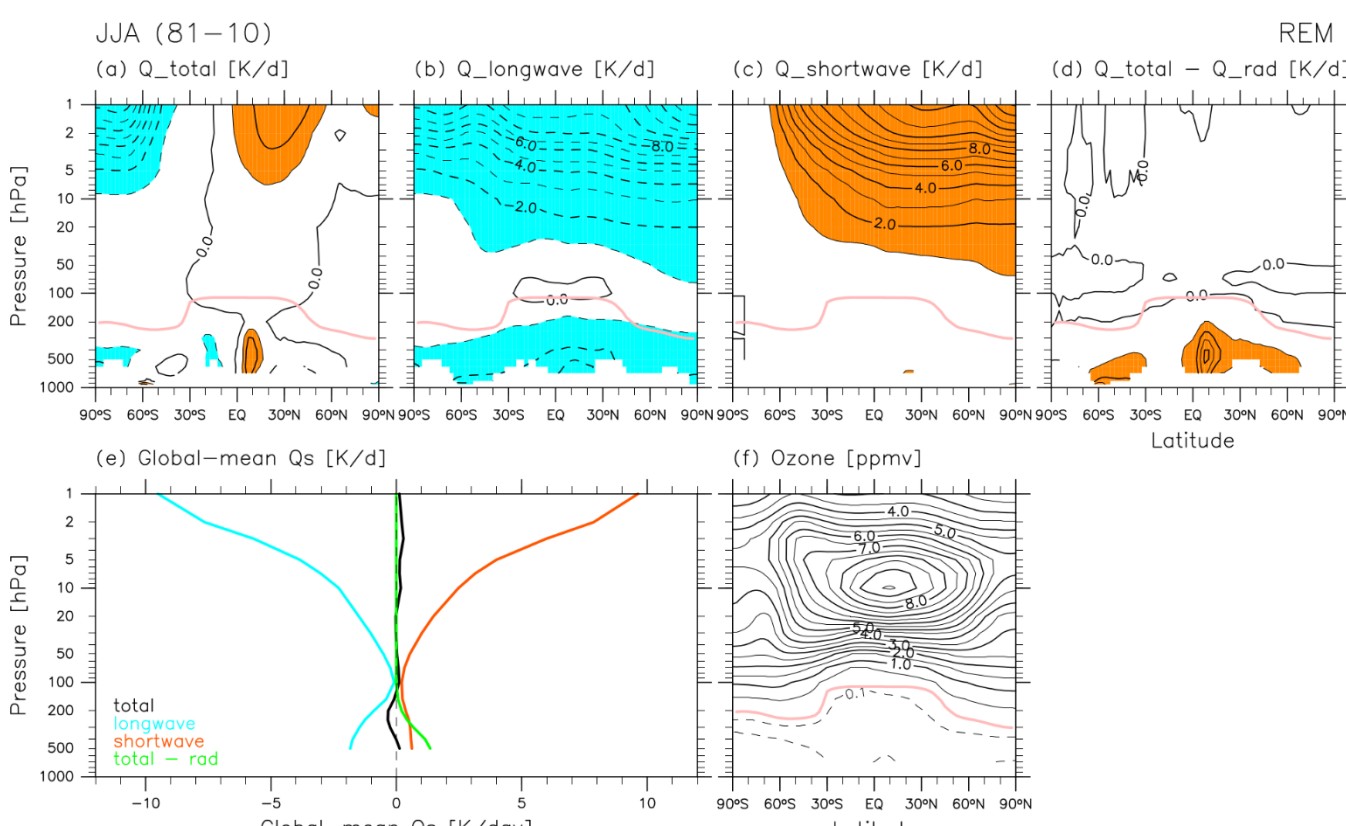

**Figure 13: As for Figure 2, but for the 30-year JJA (1981–2010) mean.**



Figure 14 shows the REM climatological distributions of all terms in the TEM momentum equation for JJA. As for DJF (Figure 3), the major terms in the stratosphere at monthly time scales are the Coriolis term and the EP flux divergence term, but with strong signals extending much higher in the SH than in the NH during JJA because of the existence of polar night jet in the SH. We notice that in the SH polar upper stratosphere, there are positive signals in EP flux divergence (due to resolved waves), which are not well balanced with the Coriolis term but are rather balanced with the residual term $\overline{\varepsilon_u}$ part of

which is due to unresolved forcing from gravity waves. This signature is found in all four reanalyses (see the Supplement Folder 2). This pattern may correspond to the one shown in a work using a high-top, high-resolution model by Watanabe et al. (2008, their Figure 9) who showed strong EP flux convergence due to gravity waves and EP flux divergence due to planetary waves in the SH high-latitude, high-altitude region for July. Around the SH midlatitude tropopause, signals in the EP flux divergence due to resolved waves correspond to the northward flow (see both Figs. 12(d) and 14(b); Birner et al.,

2013). Finally, negative signals in $\overline{\varepsilon_u}$ in the midlatitude lower stratosphere in both hemispheres, as also found for DJF (Fig. 3(f)), may result in part from unresolved forcing due to gravity waves generated by the subtropical jets (e.g. Kawatani et al., 2004; Plougonven and Snyder, 2007) and the orography (Kuchar et al., 2020). (Again, see also Podglajen et al., 2020 for a comparison of reanalyses with long-duration balloon observations.)





**Figure 14: As for Figure 3, but for the 30-year JJA (1981–2010) mean.**





Figure 15 shows REM climatological distributions of all terms in the TEM thermodynamic equation during JJA. As for DJF
(Figure 4), the major terms in the stratosphere at monthly time scales are the vertical advection term and the total (actually
radiative) diabatic heating term, but other terms show noticeable contributions at higher latitudes in the middle to upper
stratosphere. Furthermore, values of the residual term $\overline{\varepsilon_\theta}$ are on the same order of magnitude as those for the two major
terms in the upper stratosphere during JJA, indicating large differences between the observations and the forecast models in
the upper stratosphere also during JJA.




Figure 15: As for Figure 4, but for the 30-year JJA (1981–2010) mean.



### 3.2.2 Differences of each reanalysis from REM for JJA

The characteristics of the differences for temperature and zonal wind for JJA are similar to those for DJF, namely, for temperature the differences among different reanalyses become larger at higher altitudes, and for zonal wind the differences are largest in the tropics and in the low-to-midlatitude upper stratosphere (Supplement Folder 3). Figure 16 shows differences for $\Psi^*_{\bar{v}^*}$ during JJA. In the lower stratosphere below the 10 hPa level, the main, SH cell of the BD circulation (Fig. 12(g)) is overall stronger for JRA-55 and weaker for MERRA-2 and ERA-Interim (see also the differences in $\bar{v}^*$ in Supplement Folder 3). Inter-reanalysis standard deviations relative to the REM (Fig. 16(f)) indicate fractional differences of 5–50 % (i.e. larger than those for DJF) among these reanalyses in the main, SH cell of the BD circulation. The features for $\Psi^*_{\bar{v}^*}$ described above are again generally in agreement with those for $\Psi^*_{\bar{\omega}^*}$ (Supplement Folder 3).

Figure 17 shows differences in LW radiative heating during JJA. The difference patterns are similar to those for DJF (Figure 6), namely, greatest absolute differences are again found in the upper stratosphere (fractional differences of 5–10 % generally and >10 % in the winter polar stratosphere), with MERRA-2 and JRA-55 being more positive than the REM and ERA-Interim and CFSR being more negative. Furthermore, strong negative differences in CFSR and strong positive differences in JRA-55 are again consistent with the temperature differences in these two reanalyses; that is, warmer in CFSR and colder in JRA-55 (Supplement; Chapter 3 of SPARC, 2022). Thus, the LW heating differences in the upper stratosphere shown in Figure 17 may be, as for DJF (Fig. 6), largely determined by differences in temperature. LW heating differences in the troposphere and around the tropopause for JJA are also quite similar to those for DJF, and probably related mainly to differences in the distribution of clouds. Also, as for DJF (Fig. 5(f)), large fractional differences (>10 %) are found around the tropopause globally and in the tropical-to-subtropical lower stratosphere (Fig. 17(f)) where the LW heating due to $O_3$ absorption is also important. Figure 18 shows differences in SW radiative heating during JJA. Again, as for DJF, greatest absolute differences are found in the sunlit region of the upper stratosphere (fractional differences of 5–10 %), and the strong negative differences in JRA-55 are consistent with negative differences in ozone concentration in this reanalysis (Supplement; Chapter 4 of SPARC, 2022). By contrast, the differences in ozone distribution cannot fully explain the anomalies in CFSR and MERRA-2. Differences in the tropical upper troposphere during JJA are also quite similar to those during DJF, and may be related to differences in the cloud distribution (Wright et al., 2020; Chapter 8 of SPARC, 2022).





**Figure 16: As for Figure 5, but for the 30-year JJA (1981–2010) mean.**





**Figure 17: As for Figure 6, but for the 30-year JJA (1981–2010) mean.**





**Figure 18: As for Figure 7, but for the 30-year JJA (1981–2010) mean.**

570





Figures 19 and 20 show differences of each reanalysis relative to the REM during JJA for the two major terms of the TEM momentum equation. Regarding the Coriolis term (Fig. 19), characteristics in $\bar{v}^*$ (Supplement Folder 3) are reflected. In the midlatitude lower stratosphere below the 10 hPa level, JRA-55 shows generally positive differences (stronger poleward flows) and MERRA-2 shows generally negative differences (weaker poleward flow), as similar to the DJF case. Fig. 19(f) shows that inter-reanalysis fractional differences for $f\bar{v}^*$ are generally less than 50 % (larger than those for the DJF case) in the SH extratropical stratosphere and NH lower stratosphere, where strong poleward flows are found in the REM (Fig. 12(d)). In the winter hemisphere where we expect wave-driven $\bar{v}^*$, Fig. 20 shows generally negative differences in the EP flux divergence term (more convergence) in JRA-55, but more mixed results in other reanalyses compared to the DJF case. Large differences in the EP flux divergence term are found in the SH middle-to-upper stratosphere and in the extratropical lower stratosphere in both hemispheres, indicating differences in Rossby and synoptic-scale wave activity across the four reanalyses. As discussed for DJF, it is possible that the differences in the residual term may play a role for the differences in the resolved wave activity, but that the interaction between the resolved and unresolved drag differs slightly from the situation in NH during DJF (Cohen et al., 2013; Eichinger et al., 2020). Inter-reanalysis fractional standard deviations are generally less than 50 % in the extratropical stratosphere and less than 10 % in the midlatitude lower-to-middle stratosphere (Fig. 20(f)); these numbers are larger than those for the DJF case. (Also, again, see Figure 5.4 of Chapter 5 of SPARC (2022), and note the complex anomaly patterns shown in Fig. 20.)

Figures 21 and 22 show differences in each reanalysis relative to the REM for the two major terms of the TEM thermodynamic equation during JJA. As for the DJF case, differences in the vertical temperature advection term reflect inter-reanalysis differences in $\bar{\omega}^*/\bar{w}^*$ (i.e. vertical bands of anomalies; Supplement) in addition to those in temperature (with greater differences at higher latitudes; Supplement). Differences in the total diabatic heating term reflect the features in both LW and SW heatings (Figs. 17 and 18, respectively), resulting in the fact that the difference patterns in the two major terms of the TEM thermodynamic equation do not correspond well to each other. Fractional inter-reanalysis differences for both (Figs. 21(f) and 22(f)) are generally less than 50% in the regions where these terms have large positive or large negative values (Fig. 15), while those in the SH midlatitude lower-to-middle stratosphere are generally less than 10 %. Again, these results show that modern global reanalysis systems still need to improve momentum and thermodynamic balance even on the climatological zonal mean scale.





**Figure 19: As for Figure 8, but for the 30-year JJA (1981–2010) mean.**






**Figure 20: As for Figure 9, but for the 30-year JJA (1981–2010) mean.**






**Figure 21: As for Figure 10, but for the 30-year JJA (1981–2010) mean.**





**Figure 22: As for Figure 11, but for the 30-year JJA (1981–2010) mean.**



## 4 Summary

In this paper, the major variables and terms of the TEM momentum and thermodynamic equations were evaluated in the latitude-pressure domain by using four global atmospheric reanalysis data sets, MERRA-2, JRA-55, ERA-Interim, and CFSR, at climatological time scales (1980–2010) in the DJF and JJA seasons (results for MAM and SON have been shown in Supplement). The characteristics of the REM from these four reanalyses were investigated, and then the differences from the REM for each reanalysis were investigated. For the REM, variables investigated include residual vertical velocity evaluated from residual meridional velocity through the continuity equation (i.e. using the mass streamfunction), the mass

streamfunctions from both residual meridional and vertical velocities, and LW and SW radiative heatings rates. For the TEM equations, the residual terms were also calculated and investigated for their potential usefulness. The residual term for the momentum equation should include the effects of processes parameterised in the reanalysis system such as gravity waves and diffusion, effects arising from analysis increments, effects associated with using previously interpolated pressure-level data, and errors in the numerical methods (i.e. to evaluate all derivatives). The residual term for the thermodynamic equation

should include the effects of analysis increments which are the differences between the analysis state and the first guess (forecast) background state in the reanalysis system as well as effects associated with using pressure-level data and errors in the numerical methods. For the differences among different reanalyses, variables and terms primarily investigated include the mass streamfunction, LW and SW heatings, the two major terms of the TEM momentum equation (the Coriolis term and the EP flux divergence term), and the two major terms of the TEM thermodynamic equation (the vertical temperature

advection term and the total diabatic heating term).

Comparison between the original residual vertical velocity and the one estimated from residual meridional velocity revealed that the two vertical velocity fields show reasonable agreement in the troposphere and in the lower stratosphere up to 10 hPa, but differences are evident in the upper stratosphere. Because both have their own issues, looking at both estimates of

residual vertical velocity and trusting only the common features may be a good approach for studies that need very high accuracy (e.g. those on long-term trends). Comparison between the two mass streamfunctions, one calculated from residual meridional velocity and the other from residual vertical velocity, shows quantitative differences not only in the upper stratosphere (above the 10 hPa level) but also in the lower stratosphere. The diabatic heatings, in particular the LW and SW radiative heatings, from the modern global atmospheric reanalyses can be considered as the latest 'observation-based' (but

also highly model-dependent) estimates, against which those from climate models may be evaluated.

The major terms of the TEM momentum equation are the Coriolis term and the EP flux divergence term, with the latter due to the waves resolved by the reanalysis grid spacing, illustrating the wave-driven stratospheric meridional BD circulation. The residual term of the TEM momentum equation shows interesting signals in the midlatitude lower stratosphere above the

subtropical jets and in the polar upper stratosphere; they may result in part from unresolved forcing due to gravity waves





generated by the subtropical and polar night jets and the orography. The major terms of the TEM thermodynamic equation are the vertical temperature advection term and the total diabatic heating term, with the latter essentially from radiative heating. Furthermore, values of the residual term are on the same order of magnitude as those for the two major terms in the middle-to-upper stratosphere, indicating large differences between the observations and the forecast models at these altitudes.


Differences of each reanalysis from the REM and inter-reanalysis spreads for selected TEM variables were also analysed and discussed. For $\Psi^*_{\bar{v}^*}$ during DJF, the main NH cell of the BD circulation is generally stronger for JRA-55 and weaker for MERRA-2, with inter-reanalysis fractional differences of 2–10 %. For $\Psi^*_{\bar{v}^*}$ during JJA, the main SH cell of the BD circulation is generally stronger for JRA-55 and weaker for MERRA-2 and ERA-Interim, with inter-reanalysis fractional

differences of 5–50 %. For LW radiative heating during both DJF and JJA, greatest absolute differences are found in the upper stratosphere (fractional differences of 5–10 % generally and >10 % in the winter polar (upper for DJF) stratosphere), with MERRA-2 and JRA-55 being more positive, and ERA-Interim and CFSR being more negative, which may be largely determined by differences in temperature. Also, during both seasons, large fractional differences (10–50 % and even greater in some regions in particular seasons) are found around the tropopause globally and in the tropical-to-subtropical lower

stratosphere where heating due to $O_3$ absorption of upwelling LW radiation fluxes from the troposphere is also important. For SW radiative heating during both seasons, greatest absolute differences are found in the sunlit region of the upper stratosphere (fractional differences of 5–10 %), and greatest fractional differences are found in the tropical upper troposphere (>50 %) and around the extratropical tropopause (10–50 %).

Furthermore, differences for the major terms of TEM momentum and thermodynamic equations were analysed and discussed. The distribution of differences in the Coriolis term reflects that in $\bar{v}^*$. During both DJF and JJA, JRA-55 generally shows stronger poleward flows and MERRA-2 generally shows weaker poleward flows in the midlatitude lower stratosphere, with inter-reanalysis fractional differences of <10% for DJF and up to 50 % for JJA in the winter extratropical stratosphere and in the summer lower stratosphere, where strong poleward flows are found in the REM. For the EP flux divergence term, in the

winter hemisphere where we expect wave-driven $\bar{v}^*$, we found qualitatively corresponding differences to those in the Coriolis term in particular during DJF, and inter-reanalysis fractional differences in the extratropical stratosphere are generally <10% during DJF and <50 % during JJA. For the two major terms of the TEM thermodynamic equation, during both seasons, differences in the vertical temperature advection term reflect those in $\bar{\omega}^*/\bar{w}^*$ (i.e. vertical bands of anomalies) in addition to those in temperature (with greater differences at higher altitudes), while differences in the total diabatic heating

term reflect the features in both LW and SW heatings, resulting in the fact that the difference patterns in these two terms do not correspond well to each other. Fractional inter-reanalysis differences for both terms during both seasons are generally <50% in the regions where these terms have large positive or large negative values, while those in the winter midlatitude lower-to-middle stratosphere are generally less than 10 %. These results show that modern global reanalysis systems still need to improve momentum and thermodynamic balance even on the climatological zonal mean scale.

The results shown in this paper provide fundamental information on the quality of the recent global atmospheric reanalyses in the stratosphere and upper troposphere in the zonal-mean TEM framework. Our analysis indicates that the calculated residual term of the TEM momentum equation can be useful to investigate the role of gravity waves if the impact of gravity waves is greater than the impact of the reanalysis increments on the momentum balance. (Note that the role of gravity waves

for the zonal momentum budget will be more accurately constrained in more recent reanalyses, which have higher resolutions and resolve larger part of the gravity wave spectrum (e.g. Li et al, 2023; Gupta et al., 2021).) The calculated residual term of the TEM thermodynamic equation can be useful to investigate the analysis increments, highlighting the regions where the forecast models need further improvements.

**Supplement**

There are separate supplementary materials which include figures for various TEM terms and variables, for the REM, each reanalysis, and differences of each reanalysis with respect to the REM, and for 30-year DJF, MAM, JJA, and SON means.

**Data availability**

See Section 2.1 for the access information for all the data sets analysed in this paper.

**Author contributions**

PM and JSW prepared the zonal mean dynamical and thermodynamical data sets, and SMD prepared the common-grid

ozone data set. MF made all data analysis, created all the figures, and drafted the manuscript. BMS and TB contributed to additional discussions linked to Chapter 5 of SPARC (2022). All the authors contributed to interpretation and discussion of the results and to improvements of the manuscript.

**Competing interests**

Among the authors, PS and BMM are members of the editorial board of ACP, and MF and JSW are co-organizers of the ACP/WCD inter-journal special issue, "The SPARC Reanalysis Intercomparison Project (S-RIP) Phase 2".



## Acknowledgements

We acknowledge the scientific guidance and sponsorship of the World Climate Research Programme (WCRP) coordinated in the framework of Stratosphere-troposphere Processes And their Role in Climate (SPARC) and the SPARC Reanalysis Intercomparison Project and its Phase 2 (S-RIP, and S-RIP2). We thank the reanalysis centres for providing their support and data products. Masatomo Fujiwara's contribution was financially supported in part by the Japan Society for the Promotion of Science (JSPS) through grants in aid for Scientific Research (JP26287117, JP16K05548, JP18H01286, and 22H01303). Patrick Martineau acknowledges support as an international research fellow of the Japan Society for the Promotion of Science (P17029) and support also by MEXT through the ArCS Project. Jonathon Wright acknowledges funding from the National Natural Science Foundation of China (42275053). Marta Abalos acknowledges funding from the Spanish National project RecO3very (PID2021-124772OB-I00). Beatriz M. Monge-Sanz acknowledges funding from the UK Natural Environment Research Council (NERC) through the ACSIS project (North Atlantic Climate System Integrated Study) led by the National Centre for Atmospheric Science (NCAS). The NXPACK library developed by Masato Shiotani was used for handling netCDF files by Masatomo Fujiwara. The GFD-DENNOU library was used for producing Figs. 1–22 and all the figures in the Supplement. We thank Yoshihiro Tomikawa for valuable discussion.

## Financial support

This research has been supported by the Japan Society for the Promotion of Science (grant nos. JP26287117, JP16K05548, JP18H01286, 22H01303, and P17029), the National Natural Science Foundation of China (42275053), the Spanish National project RecO3very (PID2021-124772OB-I00), and the UK Natural Environment Research Council (NERC) through the ACSIS project (North Atlantic Climate System Integrated Study) led by the National Centre for Atmospheric Science (NCAS).

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
