# Peer review of "Climatology of the terms and variables of transformed Eulerian-mean (TEM) equations from multiple reanalyses: MERRA-2, JRA-55, ERA-Interim, and CFSR"

_EGUsphere, 2023_

## Referee Comment (RC1)

**Comments to the paper entitled "Climatology of the terms and variables of transformed Eulerian-mean (TEM) equations from multiple reanalyses: MERRA-2, JRA-55, ERA-Interim, and CFSR" by Fujiwara et al.**

**General comment:**

This study examines 30-year climatology of the major variables and terms of the transformed Eulerian-mean (TEM) momentum and thermodynamic equations by using four global reanalyses data including MERRA-2, JRA-55, ERA-Interim, and CFSR for boreal winter (December–February, DJF) and summer (June–August, JJA). By calculating the reanalysis ensemble mean (REM) of the individual terms in the TEM equations, the authors illustrate the climatological properties and relative importance of the terms. Through this analysis, a significant magnitude of the residual is identified in both the momentum and thermodynamic energy equations and their potential sources are also discussed. Differences in each of the four reanalysis datasets compared to the REM exhibit distinct features, indicating inconsistency among the reanalysis data in representing the dynamical structures of the troposphere and the stratosphere.

While the authors make the best effort to calculate and visualize the various terms in TEM equations with caution, 1) the sequence of analysis in this paper makes it challenging to connect specific results with their respective causes. In this regard, the differences among each reanalysis data are just listed without a comprehensive summary. 2) Insufficient elucidation regarding the causes of the differences also calls for additional clarification. Moreover, 3) despite the division of analysis into DJF and JJA, the discussion on seasonal variations appears insufficient, giving the impression that the aspects observed in winter are likewise depicted in summer. Therefore, I hope that the authors will refine the manuscript taking into account the suggested revisions, making it novel enough for publication in ACP. The specific comments are as follows.

**Major comments**

1. As the analysis alternates between the momentum and the thermodynamic equation, there appears to be a deficiency in establishing a seamless connection between the results and their underlying causes. Hence, it is recommended to commence the analysis with the momentum equation and subsequently address the thermodynamic energy equation, accompanied by a rearrangement of the figures accordingly.

2. A matter related to Major Comment 1 is observed concerning the discussion of differences among reanalysis datasets. The content addressing these distinctions appears detached and comes much later without a link, making it challenging to summarize the causes and outcomes of these differences. Examples are as follows:

    A. **Differences in the meridional circulation:**
    Regarding a stronger (weaker) residual-mean meridional circulation represented by JRA-55 (MERRA-2) compared to REM (Figure 5, L356–357), the authors attribute stronger (weaker) $\bar{v}^*$ described in Supplementary Folder 3 as a responsible cause. Since $\bar{v}^*$ is associated with the resolved wave forcing (Eq. 1), I expect the analysis of EPFD following this finding. However, the discussion about EPFD takes place in L412–418 with Figure 9 after the discussion on radiative heating. Accordingly, the fact that JRA-55 (MERRA-2) has negative (positive) EPFD differences in the dominant negative EPFD regions, which indicates that the overestimation (underestimation) of negative EPFD in comparison to REM, is not perceived to be connected to the stronger (weaker) meridional circulation in JRA-55 (MERRA-2). As Figure 8 describes the Coriolis forcing $f\bar{v}^*$, rearranging the order to present Figures 5 followed by Figure 8 and 9 could enable the authors to maintain the same explanation, while providing a comprehensive summary for the distinct meridional circulations between JRA-55 and MERRA-2.

    B. **Differences in the total radiative heating in Figure 11:**
    According to Figure 2, 6, and 7, it is identified that ERA-Interim and CFSR tend to overestimate the longwave (LW) cooling as well as shortwave (SH) warming, although the responsible cause are different. Conversely, MERRA-2 and JRA-55 tend to underestimate them. However, in Figure 11, MERRA-2 and ERA-Interim (JRA-55 and CFSR) exhibit positive (negative) deviation of the total heating from REM. Based on the findings in Figure 6 and 7, the differences in total heating shown in Figure 11 could be connected to the aforementioned tendencies with respect to LW and SW. In the case of CFSR (ERA-Interim), the overestimation of LW cooling is greater (less) than that of SW warming, contributing to the negative (positive) total heating difference. In contrast, for JRA-55 (MERRA-2), underestimation of LW cooling is less (greater) than that of SW warming, leading to negative (positive) total heating difference.

3. The authors conduct the same analyses during both winter and summer, presenting the same figures for both seasons. However, if seasonal variations do not significantly impact the results, it might be more concise and appropriate to show only the key differences in the main results and move the remaining details to the supplementary material, emphasizing the essential findings.

**Minor comments**

1. L90: The sentence is not well organized. Below sentence is one of the recommendations.
   *We first present the findings for the reanalysis ensemble mean (REM), followed by an analysis of the discrepancies of each reanalysis from the REM during DJF and JJA.*

2. L212–213: I think there is no need to separate the temperature description by altitude since the Northern Hemisphere stratosphere is consistently colder than the Southern Hemisphere stratosphere across all altitudes.

3. L215: Please specify the altitude of two maxima of the upwelling in the tropics

4. L244: reanalyses.The > reanalyses. The

5. L244: Remove the closing parenthesis at the end of this sentence.

6. L312: Podglajen et al., 2020 > Podglajen et al. (2020)

7. L351–353: Please consider adding a brief mention or acknowledgment of the observed temperature differences in the reanalyses, as the temperature plays a significant role in the radiative heating.

8. L310–315, L515–519: CFSR and MERRA-2 reanalysis data provide the parameterized orographic gravity wave drag (GWD) and the sum of orographic and non-orographic GWD, respectively. JRA-55 also offer the parameterized GWD, while the Rayleigh damping effect is also included. Therefore, it would be beneficial to analyze the contribution of these GWD to the residual as a means of validating the authors speculation.

---

## Author Comment (AC1)

**Response to comments by Reviewer #1**

Thank you very much for your valuable comments and suggestions. Please see below for our answers to yours.

**General comment:**

This study examines 30-year climatology of the major variables and terms of the transformed Eulerian-mean (TEM) momentum and thermodynamic equations by using four global reanalyses data including MERRA-2, JRA-55, ERA-Interim, and CFSR for boreal winter (December–February, DJF) and summer (June–August, JJA). By calculating the reanalysis ensemble mean (REM) of the individual terms in the TEM equations, the authors illustrate the climatological properties and relative importance of the terms. Through this analysis, a significant magnitude of the residual is identified in both the momentum and thermodynamic energy equations and their potential sources are also discussed. Differences in each of the four reanalysis datasets compared to the REM exhibit distinct features, indicating inconsistency among the reanalysis data in representing the dynamical structures of the troposphere and the stratosphere.

Thank you very much for your very nice summary of our results.

While the authors make the best effort to calculate and visualize the various terms in TEM equations with caution, 1) the sequence of analysis in this paper makes it challenging to connect specific results with their respective causes. In this regard, the differences among each reanalysis data are just listed without a comprehensive summary. 2) Insufficient elucidation regarding the causes of the differences also calls for additional clarification. Moreover, 3) despite the division of analysis into DJF and JJA, the discussion on seasonal variations appears insufficient, giving the impression that the aspects observed in winter are likewise depicted in summer. Therefore, I hope that the authors will refine the manuscript taking into account the suggested revisions, making it novel enough for publication in ACP. The specific comments are as follows.

Regarding 1), we will follow your practical suggestions which are written below.

Regarding 2), we thank the reviewer for the suggestions of additional discussion which are written below. They are very helpful.

Regarding 3), three reviewers have provided us with different suggestions on the choice of season(s).

Reviewer #1 suggested that it may be sufficient to show either DJF or JJA only in the main text and move the other to the Supplement. Reviewer #2 suggested to add annual means and/or some additional months. Reviewer #3 suggested to show either MAM or SON.

We would like to keep the current choice, i.e. showing both DJF and JJA with MAM and SON being shown in the Supplement. This is because DJF and JJA are the two contrasting seasons often discussed in tandem in the literature. Our analysis shows that the characteristics of the differences are qualitatively similar (with quantitative differences) in the winter/summer hemisphere for DJF and JJA. Therefore, if we show only the results for e.g. DJF in the main text, we expect that readers would wonder about the other season, JJA, or vice versa, whereas leaving results for the equinox seasons MAM and SON in the supplement saves space while still making these seasons available for examination.

**Major comments**

1. As the analysis alternates between the momentum and the thermodynamic equation, there appears to be a deficiency in establishing a seamless connection between the results and their underlying causes. Hence, it is recommended to commence the analysis with the momentum equation and subsequently address the thermodynamic energy equation, accompanied by a rearrangement of the figures accordingly.

Thank you for your suggestion. We will follow your advice and switch Figures 2 and 3, Figures 6-7 and 8-9, etc. in the revised manuscript.

2. A matter related to Major Comment 1 is observed concerning the discussion of differences among reanalysis datasets. The content addressing these distinctions appears detached and comes much later without a link, making it challenging to summarize the causes and outcomes of these differences. Examples are as follows:

A. Differences in the meridional circulation: Regarding a stronger (weaker) residual-mean meridional circulation represented by JRA-55 (MERRA-2) compared to REM (Figure 5, L356–357), the authors attribute stronger (weaker) $\bar{v}^*$ described in Supplementary Folder 3 as a responsible cause. Since $\bar{v}^*$ is associated with the resolved wave forcing (Eq. 1), I expect the analysis of EPFD following this finding. However, the discussion about EPFD takes place in L412–418 with Figure 9 after the discussion on radiative heating. Accordingly, the fact that JRA-55 (MERRA-2) has negative (positive) EPFD differences in the dominant negative EPFD regions, which indicates that the overestimation (underestimation) of negative EPFD in comparison to REM, is not perceived to be

connected to the stronger (weaker) meridional circulation in JRA-55 (MERRA-2). As Figure 8 describes the Coriolis forcing $f\bar{v}^*$, rearranging the order to present Figures 5 followed by Figure 8 and 9 could enable the authors to maintain the same explanation, while providing a comprehensive summary for the distinct meridional circulations between JRA-55 and MERRA-2.

Thank you very much for pointing these characteristics out. In the revised manuscript, we will change the order of figures and paragraphs in the Differences section as you suggested to improve the flow of the manuscript.

B. Differences in the total radiative heating in Figure 11: According to Figure 2, 6, and 7, it is identified that ERA-Interim and CFSR tend to overestimate the longwave (LW) cooling as well as shortwave (SH) warming, although the responsible cause are different. Conversely, MERRA-2 and JRA-55 tend to underestimate them. However, in Figure 11, MERRA-2 and ERA-Interim (JRA-55 and CFSR) exhibit positive (negative) deviation of the total heating from REM. Based on the findings in Figure 6 and 7, the differences in total heating shown in Figure 11 could be connected to the aforementioned tendencies with respect to LW and SW. In the case of CFSR (ERA-Interim), the overestimation of LW cooling is greater (less) than that of SW warming, contributing to the negative (positive) total heating difference. In contrast, for JRA-55 (MERRA-2), underestimation of LW cooling is less (greater) than that of SW warming, leading to negative (positive) total heating difference.

Thank you very much for describing all these characteristics, which will be included in the revised manuscript.

3. The authors conduct the same analyses during both winter and summer, presenting the same figures for both seasons. However, if seasonal variations do not significantly impact the results, it might be more concise and appropriate to show only the key differences in the main results and move the remaining details to the supplementary material, emphasizing the essential findings.

As explained above, we prefer to keep both DJF and JJA in the main text. In the Abstract and in Section 4 (Summary), we think that your point is clear.

**Minor comments**

1. L90: The sentence is not well organized. Below sentence is one of the recommendations.
*We first present the findings for the reanalysis ensemble mean (REM), followed by an analysis of the*

*discrepancies of each reanalysis from the REM during DJF and JJA.*

Will be considered in the revised manuscript.

2.  L212–213: I think there is no need to separate the temperature description by altitude since the Northern Hemisphere stratosphere is consistently colder than the Southern Hemisphere stratosphere across all altitudes.

Will be considered in the revised manuscript.

3.  L215: Please specify the altitude of two maxima of the upwelling in the tropics

The two small local maxima in w_res (Figure 1e) are located around 50-70 hPa (both at 50 hPa and at 70 hPa, two maxima are found at 15S and at 12.5N). The point is that the equatorial 50-70 hPa has a minimum, not a maximum.

4.  L244: reanalyses.The > reanalyses. The

A space will be added.

5.  L244: Remove the closing parenthesis at the end of this sentence.

Will be removed.

6.  L312: Podglajen et al., 2020 > Podglajen et al. (2020)

Will be corrected.

7.  L351–353: Please consider adding a brief mention or acknowledgment of the observed temperature differences in the reanalyses, as the temperature plays a significant role in the radiative heating.

Will be added.
DJF: In the upper stratosphere, JRA-55 is colder and CFSR is warmer with MERRA-2 and ERA-Interim being in the middle. (For longwave heating, JRA-55 shows positive anomalies, and CFSR shows negative anomalies; in other words, CFSR has stronger cooling, consistent with the cooling to

space theory, i.e. the Newtonian cooling approximation.)

JJA: The tendencies are the same as those in DJF.

8. L310–315, L515–519: CFSR and MERRA-2 reanalysis data provide the parameterized orographic gravity wave drag (GWD) and the sum of orographic and non-orographic GWD, respectively. JRA-55 also offer the parameterized GWD, while the Rayleigh damping effect is also included. Therefore, it would be beneficial to analyze the contribution of these GWD to the residual as a means of validating the authors speculation.

The residual here means those components that are not resolved on the common grids of the zonal mean data set, and is not directly (at least not "directly") related to e.g. the particular gravity wave parameterization used in each reanalysis system. Observational data assimilation is also a key component of the reanalysis system and governs the quality of the final reanalysis products at leading order. The GWD parameterization (please see S-RIP Final Report Chapter 2 (SPARC, 2022), Table 2.7 for the gravity wave drag parameterization used in the forecast model of each reanalysis) provides drag to the system within the forecast model part, but observational data assimilation probably still exerts the largest influence on the final data products.

Nevertheless, the reviewer is correct that all four reanalyses provide zonal acceleration (zonal wind tendencies) due to gravity wave drag parameterization and other parameterizations (please see the final 3 pages of this letter for detailed information on which data are available from each reanalysis), and looking at these data does provide some insight. Figures R1 to R4 show zonal wind tendencies due to parameterizations from the four reanalyses. These figures confirm our original speculation that, in the stratosphere, GWD plays a major role in this residual. Note that different reanalyses apply different schemes for the parameterizations (see Chapter 2 of SPARC, 2022), resulting in different distributions for e.g. GWD among different reanalyses. In particular, JRA-55 applies a Rayleigh dampling at pressures less than 50 hPa that mimics drag due to non-orographic GWD (Section 3.1of Sato and Hirano, 2019; private communication with Yayoi Harada and Chiaki Kobayashi of JMA, 2024) whose data have been added to the provided parameterized GWD data, which is, most probably, the cause of the signals in the upper stratosphere. We will add this discussion to the revised manuscript (the main text) and the figures for all four seasons in the Supplement.

[Figure]

**Figure R1. Zonal acceleration due to (a) all parameterizations, (b) parameterized gravity wave drag (including both orographic and non-orographic gravity waves), (c) moist processes, and (d) turbulence for MERRA-2 based on 30-year DJF means (1980-2010). The contours and colour shading are the same as for Figure 3.**

[Figure]

**Figure R2. Zonal acceleration due to (a) all parameterizations, (b) parameterized gravity wave drag (orographic gravity wave drag only, plus a Rayleigh damping applied at pressures less than 50 hPa that mimics drag due to non-orographic gravity waves; see Sato and Hirano (2019)), (c) convective processes, and (d) vertical diffusion for JRA-55 based on 30-year DJF means (1980-2010). The contours and colour shading are the same as for Figure 3.**

[Figure]

**Figure R3. Zonal acceleration due to all parameterizations for ERA-Interim based on 30-year DJF means (1980-**

[Figure]

**Figure R4. Zonal acceleration due to (a) all parameterizations, (b) parameterized gravity wave drag (orographic only), (c) convective mixing, and (d) vertical diffusion for CFSR based on 30-year DJF means (1980-2010). The contours and colour shading are the same as for Figure 3.**

Reference:

Sato, K. and Hirano, S.: The climatology of the Brewer–Dobson circulation and the contribution of gravity waves, Atmos. Chem. Phys., 19, 4517–4539, https://doi.org/10.5194/acp-19-4517-2019, 2019.

SPARC: SPARC Reanalysis Intercomparison Project (S-RIP) Final Report, edited by Fujiwara, M., Manney, G. L., Gray, L. J., and Wright, J. S., SPARC Report No. 10, WCRP-6/2021, 612 pp., https://doi.org/10.17874/800dee57d13, available also at https://www.sparc-climate.org/sparc-report-no-10/ (last access: 16 February 2023), 2022.

See also the next 3 pages for the detailed information on the zonal acceleration tendencies provided by each reanalysis.

**Information on zonal acceleration (zonal wind tendency) data provided by each reanalysis:**

**[1] MERRA-2**

Documentation:

https://gmao.gsfc.nasa.gov/pubs/docs/Bosilovich785.pdf

tavg3_3d_udt_Np (M2T3NPUDT): Wind Tendencies

Data files:

https://doi.org/10.5067/YSR6IA5057XX

MERRA-2 tavgM_3d_udt_Np: 3d,Monthly mean,Time-Averaged,Pressure-Level,Assimilation,Wind Tendencies V5.12.4 (M2TMNPUDT)

Relevant variables:

DUDTANA:long_name = "total_eastward_wind_analysis_tendency" ; (Not shown in the figures)

DUDTDYN:long_name = "tendency_of_eastward_wind_due_to_dynamics" ; (Not shown in the figures)

DUDTGWD:long_name = "tendency_of_eastward_wind_due_to_GWD" ;

DUDTMST:long_name = "zonal_wind_tendency_due_to_moist" ;

DUDTTRB:long_name = "tendency_of_eastward_wind_due_to_turbulence" ;

**[2] JRA-55:**

Documentation:

https://jra.kishou.go.jp/JRA-55/document/JRA-55_handbook_LL125_en.pdf

4.1.12. Isobaric average diagnostic fields (fcst_phy3m125)

Data files:

https://data.diasjp.net/dl/storages/filelist/dataset:204

--> Hist, Monthly, fcst_phy3m125

--> gwdua.

Relevant files/variables:

fcst_phy3m125_gwdua : Gravity wave zonal acceleration (*)

fcst_phy3m125_adua : Adiabatic zonal acceleration (Not shown in the figures)

fcst_phy3m125_cnvua : Convective zonal acceleration

fcst_phy3m125_vdfua : Vertical diffusion zonal acceleration

(*) Note that fcst_phy3m125_gwdua includes zonal acceleration due to both orographic GWD and Rayleigh damping. The latter is applied at all pressures less than 50 hPa and mimics non-orographic GWD. The references on this are Sato and Hirano (2019, Section 3.1) and private communication with Yayoi Harada and Chiaki Kobayashi of JMA (2024).

Reference:
Sato, K. and Hirano, S.: The climatology of the Brewer–Dobson circulation and the contribution of gravity waves, Atmos. Chem. Phys., 19, 4517–4539, https://doi.org/10.5194/acp-19-4517-2019, 2019.

**[3] ERA-Interim:**
Documentation:
https://confluence.ecmwf.int/display/CKB/ERA-Interim%3A+documentation
Table 10. Accumulated model full levels net tendencies

Relevant variables:
"u tendency" (parameter ID: 112; the original units: m s-1 (for 6-hour accumulation))

Monthly mean zonal mean data were created by Marta Abalos for the paper by Abalos et al. (JGR, 2015) based on 6-hourly model-level data above ~500 hPa.

Reference:
Abalos, M., Legras, B., Ploeger, F., and Randel, W. J.: Evaluating the advective Brewer-Dobson circulation in three reanalyses for the period 1979–2012, J. Geophys. Res., 120, 7534–7554, https://doi.org/10.1002/2015JD023182, 2015.

**[4] CFSR:**
Data files:
https://rda.ucar.edu/datasets/ds093.2/dataaccess/

Relevant variables:

Convective gravity wave drag zonal acceleration (Not shown; it was found that this variable is filled with zero or fill/missing value)

Convective zonal momentum mixing acceleration

Gravity wave drag zonal acceleration

Vertical diffusion zonal acceleration

Other settings when downloading:

Monthly Mean (4 per day) of 6-hour Average (initial+0 to initial+6)

GRID: 2.5deg. x 2.5 deg.

(Thus, the file names are diabl06.gdas.YYYYMM.grb2, where YYYYMM is e.g. 201001)

---

## Author Comment (AC2)

**Response to comments by Reviewer #2**

Thank you very much for your valuable comments and suggestions. Please see below for our answers to yours.

Review Comments for the Manuscript: "Climatology of the terms and variables of transformed Eulerian-mean (TEM) equations from multiple reanalyses: MERRA-2, JRA-55, ERA-Interim, and CFSR."

General Comments:

The manuscript presents an in-depth analysis of the principal variables and terms of the TEM momentum and thermodynamic equations, utilizing datasets spanning over thirty years from MERRA-2, JRA-55, ERA-Interim, and CFSR. The detailed scrutiny of the reanalysis ensemble mean (REM), alongside the notable discrepancies among individual reanalyses, substantially enriches our understanding of atmospheric dynamics and radiative equilibrium from the troposphere to the mesosphere. The study's potential to improve reanalysis datasets and enhance atmospheric modeling and simulation techniques is highly commendable and noteworthy.

Thank you very much for your full understanding of the nature of this manuscript.

Specific Comments:

1. Seasonal Tendency Analysis:

- The methodology involving DJF and JJA to analyze winter and summer tendencies in the northern hemisphere mainly captures transitions from December 1st to the end of February. This approach may only partially encompass the climatic variations in temperature, wind fields, and stream functions among the different reanalyses throughout the year. Notably, the TEM momentum equation analysis focuses on the differences between states separated by three months rather than the formation of an average state over an entire season. It is recommended to broaden the analysis to include the whole of the annual climatic mean or to analyze TEM terms for additional months. This could provide a more detailed understanding of the discrepancies among the datasets and their underlying causes.

First, please note that we provide the results for MAM and SON as well in the Supplement. We

chose to show and discuss only the DJF and JJA results in the main text because DJF and JJA are the two contrasting seasons often discussed in the literature. In this paper, we intend to show and discuss climatological means, and we decided to show 3-month climatological means as the first element of a full climatological analysis. We agree with the reviewer that there is no perfect definition for a climatology, but we believe that this choice will be useful for many colleagues.

- The need for more analysis for shorter time cycles, such as monthly budgets, is evident. For instance, events like Sudden Stratospheric Warming (SSW), typically accompanied by significant planetary wave activity, may take place within these three months and might resolve before the end of February. Consequently, the monthly variations in atmospheric momentum and thermodynamics and their causes still need to be addressed.

We could provide monthly climatologies rather than 3-month climatologies, but please note that we present 30-year averages to reduce the impacts of internal variability, including SSW events. In this manuscript, we intend to show and discuss the seasonal background states against which short-term fluctuations and events occur. Again, we agree with the reviewer that there are other valid and useful ways to define climatologies, but we believe that 3-month 30-year climatologies will be useful for many colleagues. Analysis of the TEM budget during SSW is beyond the scope of the present study, and it would involve grouping of data around the central date of the SSW, not simply considering monthly mean, and could constitute a separate paper, in line with Martineau et al. (2018).

Reference:
Martineau, P., Son, S.-W., Taguchi, M., and Butler, A. H.: A comparison of the momentum budget in reanalysis datasets during sudden stratospheric warming events, Atmos. Chem. Phys., 18, 7169–7187, https://doi.org/10.5194/acp-18-7169-2018, 2018.

1. Attribution of Differences in TEM Thermodynamic Terms:

- The study ascribes specific TEM thermodynamic terms variances to differences in parameters such as ozone and temperature across the datasets. A more detailed discussion and analysis of these parameters, especially ozone distribution, is advised to reinforce this attribution. This would solidify the argument and provide a more transparent explanation of the observed discrepancies, potentially elucidating the underlying mechanisms involved.

Thank you for this suggestion. We will add more detailed discussion including those suggested by Reviewer #1. Temperature differences in part explain the differences in longwave heating as

described in the response letter to Reviewer #1. Yes, ozone differences also contribute to the differences in shortwave heating, although we should note that the ERA-Interim ozone products analyzed here were not used for radiative transfer calculations in the ERA-Interim forecast model (an independent climatological ozone distribution was used instead), making interpretation of these differences a little bit complicated. However, in general, JRA-55 has less ozone in the middle to upper stratosphere than MERRA-2 and CFSR in both DJF and JJA. It is therefore consistent that JRA-55 has the minimum heating there among the three reanalyses. Differences in ozone and shortwave heating between MERRA-2 and CFSR are more difficult to explain from this perspective alone, suggesting that other factors in the radiative schemes also play a role. These two forecast models use different broadband models for both shortwave and longwave, and make different assumptions for the prescribed distributions of radiatively active gases (see Chapter 2 of SPARC, 2022), both of which will impact the stratospheric radiative equilibrium in ways that are difficult to untangle.

Reference:

SPARC: SPARC Reanalysis Intercomparison Project (S-RIP) Final Report, edited by Fujiwara, M., Manney, G. L., Gray, L. J., and Wright, J. S., SPARC Report No. 10, WCRP-6/2021, 612 pp., https://doi.org/10.17874/800dee57d13, available also at https://www.sparc-climate.org/sparc-report-no-10/ (last access: 16 February 2023), 2022.

Technical Corrections:

- Line 305: Clarification is needed on how the equatorward flow is observed from Figures 1(d) or Figure 4(b).

Figure 1(d) will be corrected as Figure 1(e). The signals we are referring to are located around 40°N and 200 hPa and around 50-60°S and 20-300 hPa, respectively. We will add this information in the text.

In conclusion, the manuscript significantly contributes to the fi eld of atmospheric sciences. I think addressing the points mentioned above could significantly enhance the depth and impact of your study. I eagerly anticipate the revised manuscript and am optimistic about the potential of this research to advance our understanding of atmospheric dynamics and modeling.

Again, thank you very much for your full understanding of the nature of the manuscript.

---

## Author Comment (AC3)

**Response to comments by Reviewer #3**

Thank you very much for your valuable comments and suggestions. Please see below for our answers to yours.

In this paper, a comparison is made on estimates of the transformed Eulerian mean momentum and thermodynamic budget terms from 4 reanalysis datasets: MERRA-2, JRA-55, ERA-Interim, and CSFR. Their results clearly show differences amongst these datasets. My main concerns on this manuscript are:

1.  Presenting the difference magnitudes and large uncertainties shouldn't be the highlight. The manuscript needs to explain what these differences mean in terms of the physics of the atmosphere for the highlights to be suitable for publication in a journal like Atmospheric Chemistry and Physics. Are the differences enough to suggest that some of the reanalysis suggests a significantly different form of dynamics is occurring? There may also be instances when the difference values may simply be just noise. The noise-signals and physical signals need to be clearly pointed out. Once the authors focus more on what differences actually signify crucial Physics differences amongst the reanalysis datasets, they may be able to improve the organization of the manuscript.

We believe that showing the uncertainty range for TEM variables and terms from multiple reanalyses is very important, and furthermore that these estimates are fundamental to both the current prevailing practice in analyzing these data products and our ability to interpret such analyses. We explain in further detail below.

The reanalysis system consists of a forecast model, assimilation scheme, and assimilated observational data. For the dynamical core of the forecast model, we believe that all four reanalyses use good, reasonable models. Choices of particular sub-gridscale parameterizations differs among different reanalyses, as summarized in Chapter 2 of the S-RIP Final Report (SPARC, 2022) for these four and other reanalyses (e.g. radiative transfer schemes in Table 2.4, convective parameterizations in Table 2.6, and gravity wave drag parameterizations in Table 2.7). We believe that the four reanalyses analyzed in this manuscript all make reasonable choices for these parameterizations (please refer to the response letter to Reviewer #1 where we show zonal acceleration data for these four reanalyses.) These systems are complex, and it is sometimes but not always possible to attribute particular anomalies to the use of a particular parameterization. A good example is provided by MERRA-2's treatment of gravity wave drag. In MERRA-2, an increased latitudinal profile of the gravity wave drag background source at tropical latitudes and increased intermittency are applied to ensure that the

forecast model can produce a spontaneous QBO. However, this treatment resulted in unrealistic tropical zonal winds in the stratosphere in the 1980s (Figure 2 of Kawatani et al., 2016), when data assimilation constraints were weaker than in the 2000s.

Furthermore, the final reanalysis data products are largely determined not by particular choices in the forecast models, but rather by the observational data assimilation. In some cases, particular parameter settings in the assimilation scheme can result in obvious biases in the reanalysis products. Good examples include the CFSR QBO issue (Saha et al., 2010, see the section on "QBO problem in the GSI"), the near-zero and sometimes even negative values of water vapour at and above the tropopause in CFSR (Davis et al. 2017, Wright et al. 2020), and the ERA5 vs. ERA5.1 issue (Simmons et al., 2020). Some of these issues have been noticed and solved by re-running the reanalysis systems for particular periods, but smaller issues are often not corrected or even noticed by the reanalysis centres. Some of the differences shown in the manuscript may emerge from these kinds of issues, but as data users it is practically impossible to attribute these issues unequivocally without conducting parameter-perturbation experiments, which is naturally the province of the data producers. Our role is instead to identify and highlight the issues so that they are more likely to be attributed and addressed in future development.

In addition, we do not have "reference" observations for each of the TEM terms and variables, and we therefore must rely on reanalyses for these terms and variables. Uncertainty ranges obtained from multiple recent reanalyses are thus important for evaluating and especially quantifying our current understanding of the atmosphere from the TEM point of view.

References:

Davis, S. M., Hegglin, M. I., Fujiwara, M., Dragani, R., Harada, Y., Kobayashi, C., Long, C., Manney, G. L., Nash, E. R., Potter, G. L., Tegtmeier, S., Wang, T., Wargan, K., and Wright, J. S.: Assessment of upper tropospheric and stratospheric water vapor and ozone in reanalyses as part of S-RIP, Atmos. Chem. Phys., 17, 12743–12778, https://doi.org/10.5194/acp-17-12743-2017, 2017.

Kawatani, Y., Hamilton, K., Miyazaki, K., Fujiwara, M., and Anstey, J. A.: Representation of the tropical stratospheric zonal wind in global atmospheric reanalyses, Atmos. Chem. Phys., 16, 6681–6699, https://doi.org/10.5194/acp-16-6681-2016, 2016.

Saha, S., Moorthi, S., Pan, H.-L., Wu, X., Wang, J., Nadiga, S., Tripp, P., Kistler, R., Woollen, J.,

Behringer, D., Liu, H., Stokes, D., Grumbine, R., Gayno, G., Hou, Y.-T., Chuang, H., Juang, H.-M. H., Sela, J., Iredell, M., Treadon, R., Kleist, D., Delst, P. V., Keyser, D., Derber, J., Ek, M., Meng, J., Wei, H., Yang, R., Lord, S., van den Dool, H., Kumar, A., Wang, W., Long, C., Chelliah, M., Xue, Y., Huang, B., Schemm, J.-K., Ebisuzaki, W., Lin, R., Xie, P., Chen, M., Zhou, S., Higgins, W., Zou, C.-Z., Liu, Q., Chen, Y., Han, Y., Cucurull, L., Reynolds, R. W., Rutledge, G., and Goldberg, M.: The NCEP climate forecast system reanalysis, B. Am. Meteorol. Soc., 91, 1015–1057, https://doi.org/10.1175/2010BAMS3001.1, 2010.

Simmons, A., Soci, C., Nicolas, J., Bell, B., Berrisford, P., Dragani, R., Flemming, J., Haimberger, L., Healy, S., Hersbach, H., Horányi, A., Inness, A., Muñoz-Sabater, J., Radu, R., and Schepers, D.: Global stratospheric temperature bias and other stratospheric aspects of ERA5 and ERA5.1, ECMWF Technical Memoranda, 859, 38 pp., https://doi.org/10.21957/rcxqfmg0, 2020.

SPARC: SPARC Reanalysis Intercomparison Project (S-RIP) Final Report, edited by Fujiwara, M., Manney, G. L., Gray, L. J., and Wright, J. S., SPARC Report No. 10, WCRP-6/2021, 612 pp., https://doi.org/10.17874/800dee57d13, available also at https://www.sparc-climate.org/sparc-report-no-10/ (last access: 16 February 2023), 2022.

Wright, J. S., Sun, X., Konopka, P., Krüger, K., Legras, B., Molod, A. M., Tegtmeier, S., Zhang, G. J., and Zhao, X.: Differences in tropical high clouds among reanalyses: origins and radiative impacts, Atmos. Chem. Phys., 20, 8989–9030, https://doi.org/10.5194/acp-20-8989-2020, 2020.

2.   In describing the reanalysis datasets somewhere in the methodology, more needs to be said on the differences in the physics that each model is known to already exhibit. These need to be described in a way that would help readers already get an idea of potential differences amongst the model.

We will add an overview of the reanalysis systems written above to Section 1.

3.   The results spend too much time describing dynamics that are already well-known. For example, the first sub-section describing REM means may be reduced to solely focus on the issues regarding the calculation of v* or w*. The results need to be re-written in a way that immediately focuses on the physical and/or unphysical differences amongst the reanalysis with one another and/or with REM.

We will shorten the description of the REM results, but would like to retain the relevant figure panels

(i.e. for temperature and zonal wind) and thus some text. This is because the REM results are needed for part of the interpretation later. Also, we believe that these panels provide a helpful reference for future studies, as we have already found them useful in this context in our own work.

4.   Showing DJF dynamics without mentioning sudden stratospheric warming dynamics isn't a good idea. Your methodology indicates the calculations uses monthly-mean. To mention SSWs requires the use of, at least, daily-mean datasets. You can choose to do this, or instead show one equinox season.

First, in this manuscript, we intend to show and discuss the seasonal background states against which various short-term fluctuations and events occur, including SSWs. We believe that the DJF climatology will be important and useful for further studies including those on SSWs.

Regarding SSWs, Chapter 6 of the S-RIP Final Report (SPARC, 2022; see also references therein) extensively investigated SSWs across multiple reanalyses. In particular, an intercomparison of the momentum budget in reanalysis products during SSW events was conducted by Martineau et al. (2018a), who also compared temperature and meridional heat flux. Also, Martineau et al. (2018b) showed the thermodynamic budget for a particular SSW event.

Please note that results for MAM and SON are shown in the Supplement. We chose to show DJF and JJA in the main text because DJF and JJA are the two contrasting seasons often discussed in the literature.

References:

Martineau, P., Son, S.-W., Taguchi, M., and Butler, A. H.: A comparison of the momentum budget in reanalysis datasets during sudden stratospheric warming events, Atmos. Chem. Phys., 18, 7169–7187, https://doi.org/10.5194/acp-18-7169-2018, 2018a.

Martineau, P., Wright, J. S., Zhu, N., and Fujiwara, M.: Zonal-mean data set of global atmospheric reanalyses on pressure levels, Earth Syst. Sci. Data, 10, 1925–1941, https://doi.org/10.5194/essd-10-1925-2018, 2018b.

I recommend major revisions for this manuscript.

We hope that the above explanation has clarified the key points of our manuscript.